



# Coupling of soil carbon and water cycles in two agroforestry systems in Malawi

Svenja Hoffmeister[1], Sibylle Kathrin Hassler[1,2], Friederike Lang[3], Rebekka Maier[3], Betserai Isaac Nyoka[4], and Erwin Zehe[1]

[1]Chair of Hydrology, Institute of Water and Environment, Karlsruhe Institute of Technology, Karlsruhe, 76131, Germany
[2]Institute for Meteorology and Climate Research, Atmospheric Trace Gases and Remote Sensing, Karlsruhe Institute of Technology, Eggenstein-Leopoldshafen, 76344, Germany
[3]Chair of Soil Ecology, Institute of Forest Sciences, University of Freiburg, Freiburg, 79085, Germany
[4]World Agroforestry (ICRAF), Chitedze Agricultural Research Station, Off-Mchinji Road, Lilongwe, Malawi

**Correspondence:** Svenja Hoffmeister (svenjahoffmeister@gmail.com)

**Abstract.** Consequences of climate change are likely to pose severe challenges on agriculture in Southern Africa. Agroforestry systems (AFSs) can potentially alleviate some of the adverse effects and offer adaptation solutions to a sustainable land use. Positive effects of AFSs which have been shown include increasing soil carbon (C) and nitrogen concentrations, sustaining favourable nutrient cycling, protection against erosion and increased carbon sequestration. The influence of the AFS tree com-
ponent on the water cycling of the crops, however, is still relatively unknown.

In this study we assessed the influence of gliricidia-maize intercropping on carbon cycling and water fluxes compared to maize as a sole crop at two well-established long-term experiments in central and southern Malawi, run by the World Agroforestry (ICRAF). The controlled setup and different durations of the experiments (>10 and >30 years) at the two sites provided information regarding soil-specific impacts of gliricidia on water dynamics. We examined soil C contents and density fractionation
as proxy for organic matter stability, soil physical and soil hydrological characteristics. We also monitored soil moisture and matric potential in different depths, determined retention curves on samples in the lab and from field data and analysed soil moisture responses to rainfall events to assess the influence of the AFS on water fluxes.

Our results show a clear increase in C contents and stability as a result of the gliricidia impact compared to the control, especially pronounced at the site with the generally lower baseline C contents. The treatment effect is also visible in soil physical
characteristics such as porosity and bulk density, which is mirrored by a greater saturated hydraulic conductivity. These treatment effects were, however, not directly translatable into soil water dynamics as the latter were influenced by several additional factors such as soil texture and interception. The gliricidia plots showed greater soil water storage capacities and retained overall more water, while generally both treatments were not under severe water stress during the observation period. We also noticed a protective effect against soil drying below the topsoil facilitated by the gliricidia. Furthermore, infiltration shifted
towards more immediate/macropore infiltration under gliricidia.

We conclude that the AFS treatment of adding gliricidia into maize cultivation has a considerable effect on nutrient and water cycling in the crop, while the effect on water fluxes is not straightforward. While the differences in soil moisture and matric potential never lead to a shortage of water for the crops, a detailed examination of water fluxes require respective measurement





in the field as they cannot be deduced from soil physical characteristics directly. AFS can thus not only support carbon accu-
mulation and stabilization, but a sensible combination of trees and crops can also be beneficial for a more sustainable use of
the available water.

# 1  Introduction

Agriculture in Southern Africa faces increasing challenges as a consequence of climate change (Fauchereau et al., 2003), such
as nutrient depletion (Mbow et al., 2019), soil erosion (Montanarella et al., 2016; Olsson et al., 2019) and shifts in water
availability (Trisos et al., 2022). This affects both food security (e.g. (Tumushabe, 2018; Programme, 2019)) and ecosystem
resilience (Sheppard et al., 2020; Jose, 2009), hence adapted land use systems are required to respond to these challenges.
Agroforestry systems (AFSs) – combinations of crops and/or livestock with a woody perennial species – have been shown to
offer promising management options in the light of these challenges. Depending on the species combination and the respective
climatic and land use setting, introducing a woody component into agricultural systems can provide a variety of benefits
(Sheppard et al., 2020). As shelterbelts or windbreaks, they can protect soils from erosion and crops from damages as well as
excessive evapotranspiration (Makate et al., 2019; Littmann and Veste, 2008; Cleugh, 1998). The trees within AFSs also affect
the nutrient cycling of the crops in various ways. While possible competition for nutrients might be a fear of many farmers
when first implementing AFS, many beneficial short- and long-term interactions are well documented (Akinnifesi et al., 2006).
The trees in AFS provide additional carbon (C) input and may increased soil organic matter (SOM) contents (Jose, 2009;
Kuyah et al., 2019) and stability (Maier et al., 2023), augmenting soil fertility. The input can occur in various ways, e.g. via
active incorporation of biomass into the soil, around the crops or simply the decomposition of plant materials and roots directly
around the trees (Shi et al., 2018). Nitrogen (N) deficiency is a major problem in tropical cropping systems and often the most
limiting nutrient (Ikerra et al., 1999). The integration of legume species can therefore increase plant-available N in the soil
and lead to a fertilisation effect, thus reducing the need for artificial addition of fertiliser. *Gliricidia sepium* (gliricidia) is a
well recommended intercrop legume tree species for nutrient demanding maize cultivation (Kwesiga et al., 2003). Numerous
studies have demonstrated its beneficial impact on carbon sequestration and nutrient supply when intercropped with maize
(Akinnifesi et al., 2010) while increasing yields (Ribeiro-Barros et al., 2018; De Schutter, 2012; Akinnifesi et al., 2006; Beedy
et al., 2010).
However, long-term sustainable improvement of SOM contents requires not only input but also stabilization of organic matter
at mineral soil surfaces and within aggregates. This stabilization process is dependent on soil texture (Schweizer et al., 2021)
and other physiochemical parameters like pH value, calcium (Ca) and magnesium (Mg) and the presence of sesquioxides
(Rasmussen et al., 2018). Iron (Fe-) and aluminium (Al-) containing sesquioxides are known to be important for aggregate
formation and organic matter stabilization. This is especially true in soils with low activity clays (Barthès et al., 2008), where a
relatively high ratio of pedogenic Fe to Al clay has been shown to support organic carbon stability against oxidation, allowing
persistence in the soil (Kirsten et al., 2021). Depending on how stable C is bound within the soil, it can be grouped into different
fractions: free light fraction (fLF), in aggregates occluded light fraction (oLF) and heavy fraction (HF), where C is bound to





soil minerals (Golchin et al., 1997).

The amount and stability of C in soils is strongly linked to soil structure (Bronick and Lal, 2005; Lal, 2020), and intercropping AFS with legume trees has been shown to increase aggregation (Blair et al., 2006) and aggregate stability (Chaplot and Cooper,

2015). Changes in soil structure in turn influence soil physical and hydrological characteristics, e.g. pore size distribution or water retention, affecting soil water at the macro scale (Nimmo and Akstin, 1988; Pachepsky and Rawls, 2003; Williams et al., 1983). Consequently, in this study we examined the link between C inputs and soil hydrology.

Adding organic C in the form of manure can enhance soil structuring and lead to increased total porosity and water retention of the soil (Bodner et al., 2015). This effect seems to be stronger in soils with lower initial C content (Rawls et al., 2004),

pointing to the interplay of mineral surfaces binding organic substances and subsequent aggregate formation. In a modelling study, Feifel et al. (2024) also found improved water retention and higher storage capacity of soils with increasing soil organic carbon (SOC) contents, however, the effect was dependent on soil texture and stronger in sandy soils. The texture-dependency of the SOC effect on water retention was also documented as part of a recent carbon-sensitive pedotransfer function approach (Bagnall et al., 2022) which also found a marked increase (about double) in plant-available water as a response to SOC input.

For AFS, the influence of C input on soil structure and water fluxes has also been studied, reporting that increasing C contents as a consequence of intercropping with legume trees lowered bulk density (García-Orenes et al., 2005) and improved water retention (Rawls et al., 2003). Farmers might be concerned to implement AFS due to a potential negative impact of the tree component on the crops, such as increased soil evaporation (Feifel et al., 2024) or competition for water in the root zone. This balance is very much dependent on the climatic conditions and the combination of the species in the AFS. A more detailed

discussion of the different benefits and concerns in implementing AFS can be found in the review by Sheppard et al. (2020). For instance, the different leaf stages and rooting depths together with a seasonality in rainfall did not lead to any competition for water in intercropping systems of pruned gliricidia and maize in a very common AFS in southern Malawi (Chirwa et al., 2007). However, there is still a lack of knowledge to what extent C-induced short- and long-term changes in soil structure affect water fluxes in these systems.

Effects of land-use change only become evident after several years and are needed to be observed over multidecadal time spans (Dearing et al., 2010). Hence, long-term monitoring sites or experiments are essential to evaluate the effect of C-input strategies. The World Agroforestry (ICRAF) in Malawi has been running experimental trials of combining maize crops with gliricidia plants at two sites for more than 30 and more than 10 years, respectively. The sites are located in the same climatic zone and have large overlap in the intercropping treatments included in the experiments. Additionally, management conditions

are similar (same pruning frequency, seeds, weeding activities) and also spacing between maize and gliricidia are the same. They are therefore ideal to study the effect of gliricidia on various processes and characteristics.

gliricidia is well suited for the Southern African region due to its high leaf-N content, high biomass production and drought resistance (Kerr, 2012). The gliricidia-maize intercrop has been shown to induce positive effects on soil fertility (Beedy et al., 2010), maize yields (Chirwa et al., 2003) and with the addition of small doses of inorganic fertiliser leads to higher maize

productivity (Akinnifesi et al., 2007). Maize may take advantage of the nutrients from the gliricidia prunings and gliricidia's capabilities of retrieving nutrient from deeper soil layers (Makumba et al., 2009). Further, a combination of gliricidia and maize




sequesters more soil C than a maize monoculture (Makumba et al., 2007).

The influence of the long-term SOM stabilized aggregates and other soil structure characteristics on hydrological properties of these soils has not been studied in great detail yet. Therefore, we address the following research questions:

1. Does the AFS treatment increases C contents of the soils?

2. Do changes in SOM contents and stability affect soil structure and hence hydraulic characteristics?

3. How do these changes in hydraulic characteristics manifest in soil water fluxes and state variables?

We use the unique setup of two of ICRAF's long-term experimental trials of gliricidia and maize to analyze the changes in SOM and nutrient status as well as soil hydrological properties, time series of hydrological fluxes and rainfall events.

## 2  Methods

### 2.1  Site description and management

The study was conducted at two field sites in Malawi, at Chitedze Agricultural Research station close to Lilongwe in central Malawi and Makoka Agricultural Research Station close to Zomba in the south of the country (Fig. 1). Both sites offer controlled agroforestry experiments run by the World Agroforestry, among them trials of intercropping *gliricidia sepium* (Jacq.)

Walp. trees with maize (*Zea mays* L.) (Akinnifesi et al., 2007; Chirwa et al., 2007), but differ in the duration of the experiment and in some soil and climatic conditions.

The Chitedze site (13°59'S and 33°38'E, 1146 m asl) is located in the Lilongwe plain and has a mean annual temperature of 20°C and annual rainfall amounts to 800-900 mm with 85 % of these rains occurring between November and April (Malunga et al., 2017). Soils at the site are Chromic Luvisols (IUSS Working Group, 2014; Malunga et al., 2017), providing high cation

exchange capacity and high agricultural potential for annual crops (Brown and Young, 1965). The experiment at Chitedze was established in 2009 in a split-plot design as an agroforestry demonstration site. The two parts within a plot include two different cropping practices for the maize. The plots themselves represent different combinations of maize, tree component and fertiliser application. For our experiments, only the gliricidia-maize intercropping and the control treatments were of importance.

The Makoka site (15°30'S and 35°15'E, 1029 m asl) is located in the southern region of Malawi and experiences a mean daily

temperature between 16 and 24°C (Chirwa et al., 2003) and mean annual rainfall of 1024 mm, unimodally occurring mainly between November and April (Ikerra et al., 1999; Malunga et al., 2017). The soils were classified as Ferric Lixisols (IUSS Working Group WRB, 2022; Ikerra et al., 1999). The experiment in Makoka was established in 1992 in a randomised complete block design with three replicates per treatment, treatments varying in combinations of tree component and different crops as well as fertiliser application rates (Ikerra et al., 1999). This study focused on the comparison of maize-only control plots and

the AFS plots of gliricidia with maize (Akinnifesi et al., 2007).

The management of these plots follows common practice of planting the maize in ridges each year before the rainy season starts. In the agroforestry treatments, gliricidia was planted on alternate ridges with a spacing of 1.50 m between ridges and





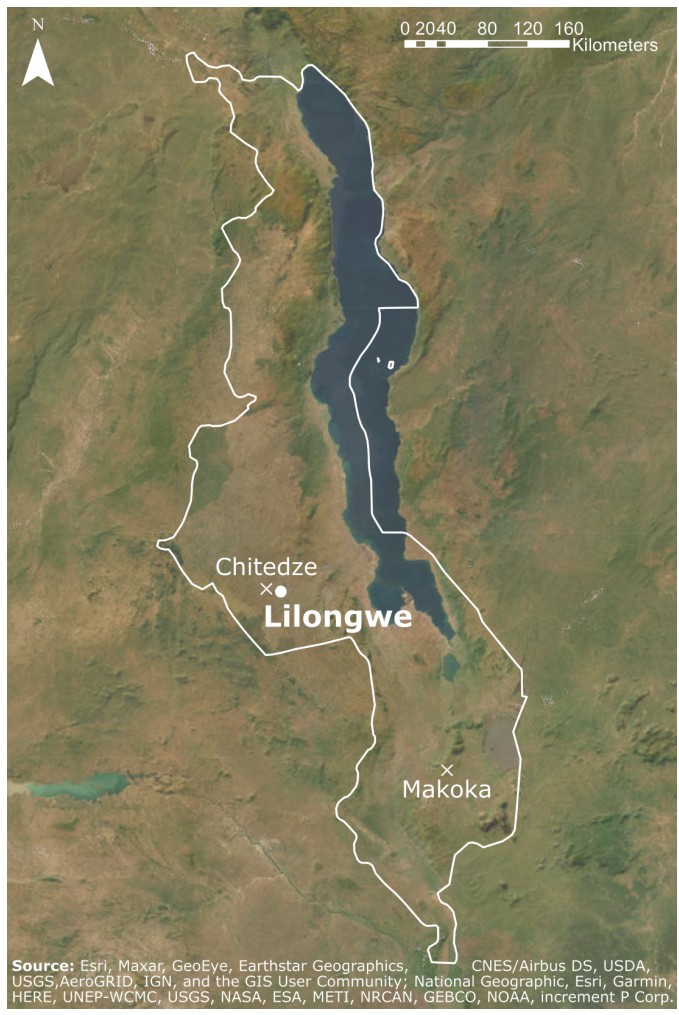

**Figure 1.** Map of the study sites Chitzedze and Makoka Research station in Malawi.

0.90 m within the ridge, resulting in a planting density of about 7,400 trees per hectare, as outlined in a previous study by Makumba et al. (2006). The gliricidia trees are cut three times a year to a height of 30 cm, in October, December, and February

and tree pruning (leaves and tender branches) are incorporated into the soil ridges. Therefore, the soil in the ridges cannot be considered as undisturbed as it is strongly mixed between the different depths.

The size of the maize plants was recorded exemplarily around the locations where we took the soil samples for measuring hydrological characteristics (big cylinders), at both sites and in both control and gliricidia plots. In Chitedze, we measured only 5 plants due to time constraints (2 in the control, 3 in the gliricidia treatment), in Makoka we measured 22 plants (10 in the

control, 12 in the gliricidia plot). Hence, this can only give a rough estimate on plant sizes. We also took photographs for visual comparison.





Pedological and hydrological characteristics were sampled and monitored at both sites in a field campaign in February/March 2022 but also including some analyses of previous campaigns in 2019 at the Chitedze site and in 2021 at Makoka site (Hoffmeister et al., 2025).

## 2.2 Soil sampling with an auger and small cylinders for analyses of texture, nutrients, soil physical and hydrological characteristics


On each of the plots we took auger samples for the analysis of nutrients, pedogenic oxides, and soil texture. We took undisturbed samples with small cylinders (4 cm height and a volume of 100 cm$^3$) for the analysis of bulk density, and (for the Makoka site only) for porosity and saturated hydraulic conductivity.

At Chitedze, the sampling was undertaken in June 2019 as part of a different study. Ten sampling points were chosen per plot within the maize ridges at a spacing of approximately 3.5 m. The auger samples were taken at each point in 0-10 and 10-20 cm depth and analysed individually for C, N, CEC and pedogenic oxides. The two depths were mixed and analysed together as a 0-20 cm sample for the density fractions and texture. Two undisturbed samples with the small cylinders were extracted at each sampling point, one for each of the two depths between 0-10 and 10-20 cm; these are in the following named 5 and 15 cm
depth.

At the Makoka site, sampling was done with only five sampling points per plot. Samples were taken within the maize ridges, similar to the sampling at Chitedze, at approximately 5 and 15 cm depth. Auger sampling was done in a campaign in 2021, pooled for 0-20 cm depth, for nutrients, pedogenic oxide, density fractions, water dispersible clay and texture. Sampling for soil physical characteristics (bulk density, porosity and saturated hydraulic conductivity) using small cylinders was done in the
main campaign in February/March 2022.

The samples were transported to Germany and analysed in the laboratory of the Chair of Soil Ecology at the University of Freiburg. We analysed texture by sieving and sedimentation after the destruction of the organic matter according to DIN ISO 11277:2002 (2002). The amount of water dispersible clay (WDC) was determined (for the samples from the Makoka site) by using only distilled water as dispersant. We estimated total exchangeable Ca-, Mg- and K-cations according to the standard
method (Anderson and Ingram, 1993) and pedogenic oxides (Fe, Al and Mn) using the dithionite extraction method (Mehra and Jackson, 1958).

Analyses of total carbon and nitrogen were done in an elemental analyser (Vario EL cube, Elementar, Langeselbold, Germany) after standard preparation techniques. In order to determine the stability of the C input by the gliricidia biomass, we also determined the relative proportions of C in different density fractions, the free light faction (fLF), the occluded light fraction
(oLF) and heavy fraction (HF) according to the method of Golchin et al. (1997) and the process described in Graf-Rosenfellner et al. (2016). Bulk density was determined by drying the samples at 105°C and weighing, then relating the dry mass to the cylinder volume. Total porosity at field moisture state was done using vacuum pycnometry. Saturated hydraulic conductivity was determined with the constant head method at steady-state.

Some more measurements of nutrients etc. were undertaken on the samples, however, they were not part of the analyses for





this study which aims at assessing the hydrological consequences of the agroforestry treatment. The data which are not shown
       here are part of the data publication (Hoffmeister et al., 2025).

### 2.3   Soil sampling with large cylinders for analyses of hydrological characteristics

At both sites we took samples in the control and intercropping plot using large cylinders (4 cm radius, 5 cm height, 250 cm$^3$
volume) to assess a range of characteristics such as bulk density, saturated hydraulic conductivity (Ksat) and water retention
curves. Samples were taken at three locations distributed in the plot, within the maize ridges and at two depths, 5 cm and 25 cm.
The cylinders were transported to Germany and analysed in the laboratory of the chair of hydrology at the Karlsruhe Institute
of Technology (KIT).
Bulk density was determined after drying the samples at 105°C and referring dry weight to the cylinder volume. The saturated
hydraulic conductivity of the samples was determined with the Ksat apparatus (UMS GmbH, Munich) which uses a Darcy
approach with a falling head. We measured water retention characteristics of the samples with the HYPROP apparatus (UMS
GmbH, Munich, Germany) which uses tensiometers in two depths to record water potential while measuring the weight of
the sample during evaporative drying. The apparatus is limited by the air entry point of the tensiometers to an approximate
minimum of -800 hPa, therefore we combined the method with the WP4C PotentiaMeter (Decagon Devices Inc., Pullman,
WA, USA), where lower potentials can be reached using a chilled mirror approach.
With the help of the in-build HYPROP software, water retention curves were parameterised using the van Genuchten equation
(Van Genuchten, 1980):

$$\theta(\psi) = \theta_r + \frac{\theta_s - \theta_r}{[1 + (\alpha|\psi|)^n]^{1-1/n}} \tag{1}$$

where $\theta_r$ is the residual water content in $m^3 m^{-3}$, the saturated water content is $\theta_s$ in $m^3 m^{-3}$, the scaling parameter is $\alpha$, the
absolute matric potential $\psi$ and the dimensionless shape parameter is $n$.
From the retention curves we also derived porosity (water content at pF = 0) and plant available water as the difference between
water contents at pF = 4.2 and pF = 1.8. Soil hydrological characteristics were also determined on the small cylinder samples
which were used for the nutrient and texture analyses. To measure total porosity, vacuum pycnometry at field moisture state
was used. Saturated hydraulic conductivity was determined using the falling-head method (Hartge and Horn, 2009).

### 2.4   Monitoring of meteorological and hydrological variables

At both the Chitedze and the Makoka sites we installed a small meteorological measurement station in the control plot to record
variables such as air temperature, relative humidity and precipitation. The temperature measurements seemed a bit spurious,
possibly due to a malfunctioning sensor and some issues with the shielding. We, therefore, did not use these measurements. The
precipitation was recorded with rain gauges (Chitedze: ECRN-100 Rain Gauge, Pullman, WA , USA; Makoka: Rain Collector,
Davis Instruments, Hayward, CA, USA). The recorded tipping counts were translated to mm rainfall by multiplying with a
factor of 0.2 as suggested by both manufacturers.
We assessed the influence of the site conditions and the agroforestry treatment onto the water fluxes in the soil by monitoring





soil moisture and matric potential in different depths at the two sites and on both control and gliricidia plots. During the field campaign in 2022 we installed soil moisture sensors (TDR-310H, Acclima, Meridian, USA) in the rooting zone of the maize close to the ridges in depths of 10, 15, 25 and 60 cm. The top sensors were at 10 cm in Makoka and 7 cm at Chitedze as
these were the shallowest possible depths to take an undisturbed sample for comparison. Unfortunately, there was some sensor failure in the maize plot at Chitedze and we only have the time series for the top sensor. In the same profiles where the soil moisture sensors were installed, we also inserted two matric potential sensors (MPS-2, Decagon Devices, Pullman, USA) at 10 and 25 cm depth. The sensors were connected to a data logger (YDOC ML417, YDOC, Ede, The Netherlands) and the fluxes were monitored from March to May 2022.

From the time series of soil moisture and matric potential responses to rainfall events were determined and water content changes calculated. We define a rainfall event as a segment of time during which more than 2 mm of rain were recorded, framed by periods without precipitation of minimum 6 hours. Precipitation events with less than 2 mm did not lead to significant changes in topsoil water content and were therefore not included in the analyses. These events were classified according to their amplitude, time to soil moisture response after the rainfall and depth until soil water storage changes were visible. The
soil water storage changes were calculated based on the difference of two successive soil water content measurements. The difference was then multiplied by the sensor depth increment of 0.18 m to acquire the change in storage. Additionally, we used the time series of field-measured soil moisture and matric potential to derive field retention curves in a similar way as for the values gained from the samples in the laboratory.

## 3 Results

### 3.1 Basic soil characteristics and maize heights at the sites

The texture classification at both sites showed differences between sites but also between treatments. At both sites, the sand fraction was the largest, followed by clay and lastly silt fractions (Table 1). Chitedze had less sand than Makoka and therefore qualified as clay loam and Makoka with a greater sand fraction as sandy clay loam (according to IUSS Working Group WRB (2022)).

At Chitedze, the sand and silt fractions of the control were larger than those of the AFS and vice versa for the clay fraction. At Makoka, differences in the treatments were reversed: the AFS sand and silt fractions exceeded the ones of the control and vice versa for the clay fraction.

The analysis of pedogenic oxides yielded higher values of both dithionite-extractable Fe ($Fe_d$) and Al ($Al_d$) in Chitedze compared to Makoka (see Table 1), with an average of 43.0 g kg$^{-1}$ and 44.1 g kg$^{-1}$ $Fe_d$ in the top 0-20 cm in Chitedze in control
and gliricidia treatment, respectively. In Makoka the values were 32.2 g kg$^{-1}$ in the maize-only plot and 28.3 g kg$^{-1}$ in the plot with gliricidia. $Al_d$ values averaged 6.7 g kg$^{-1}$ for the control and 7.0 g kg$^{-1}$ for the gliricidia plot in Chitedze and 3.7 g kg$^{-1}$ and 3.0 g kg$^{-1}$ for control and gliricidia in Makoka, respectively. Both sites have high levels of $Fe_d$ and $Al_d$. The ratio of $Fe_d$/$Al_d$ was higher in Chitedze (6.5 for the control, 6.3 including gliricidia) compared to Makoka (8.8 for the control, 9.3 including gliricidia). The difference between treatments is only significant for Chitedze's topsoil (significance level of 0.05).





**Table 1.** Different soil characteristics such as texture, pedogenic oxides and nutrients for both sites, separated according to site, treatment and depth. Abbreviations are: WDC – water-dispersable clay; Fed/Ald/Mnd – dithionite-extractable Fe/Al/Mn; CEC – cation exchange capacity; C – organic C; N – nitrogen. The values in parantheses are standard errors of the mean. For texture in Chitedze, we also show the calculated averages of the two depths for easier comparison with the Makoka data. The numbers behind the site indicate: y – sampling year and n – sample size.

| Site | Chitedze (y = 2019, n = 10) | | | | Makoka (y = 2021, n = 5) | |
|---|---|---|---|---|---|---|
| Treatment | Control | | Gliricidia | | Control | Gliricidia |
| Depth [cm] | 5 | 15 | 5 | 15 | 0-20 | 0-20 |
| Clay [g kg$^{-1}$] | 283 (4) | 315 (4) | 310 (9) | 293 (5) | | |
| Silt [g kg$^{-1}$] | 285 (4) | 258 (2) | 163 (7) | 191 (5) | | |
| Sand [g kg$^{-1}$] | 433 (4) | 428 (4) | 527 (15) | 570 (10) | | |
| WDC [g kg$^{-1}$] | | | | | 96.1 (4.5) | 70.4 (2.2) |
| Fe$_d$ [g kg$^{-1}$] | 43.05 (0.69) | | 44.12 (1.08) | | 32.21 (0.86) | 28.27 (0.52) |
| | 40.64 (0.49) | 45.45 (0.68) | 40.01 (0.84) | 48.23 (0.65) | | |
| Al$_d$ [g kg$^{-1}$] | 6.66 (0.11) | | 7.04 (0.26) | | 3.66 (0.15) | 3.03 (0.04) |
| | 6.43 (0.18) | 6.89 (0.09) | 5.94 (0.11) | 8.14 (0.09) | | |
| Fe$_d$/Al$_d$ | 6.46 | | 6.27 | | 8.80 | 9.33 |
| CEC [mmol$_c$ kg$^{-1}$] | 99.9 (2.5) | 93.9 (2.6) | 97.3 (1.7) | 88.1 (0.9) | 51 (2.2) | 90 (5.7) |
| C [g kg$^{-1}$] | 27.3 (0.5) | 30.1 (0.6) | 28.9 (0.5) | 31.4 (0.3) | 7.2 (0.3) | 16.4 (1.7) |
| N [g kg$^{-1}$] | 1.7 (0.04) | 1.9 (0.04) | 1.9 (0.03) | 2.0 (0.02) | 0.7 (0.0) | 1.4 (0.1) |
| C:N [ ] | 15.7 (0.1) | 16.0 (0.1) | 15.0 (0.1) | 15.6 (0.1) | 9.9 (0.2) | 11.3 (0.3) |

The carbon-to-nitrogen (C/N) ratio and its relationship with SOM fractionation also provided valuable insights into the degradability and stability of organic materials in the soil. The C/N ratio reflects the amount of carbon in organic material relative to nitrogen, serving as an indicator of SOM degradability and its susceptibility to microbial decomposition. The C/N ratio (Table 1) was on average considerably wider at the C-rich Chitedze site (15.6, averaged across depths and treatments) than in Makoka (10.6). The treatment effect was minimal in Chitedze, the maize monocrop having an average C/N ratio of 15.9, whereas the gliricidia had 15.3, with slightly higher values in the subsoil compared to the topsoil. In Makoka, the intercrop showed a wider C/N ratio of 15.4 than the maize control with a ratio of 9.9.

The maize plants in plots including gliricidia were larger than in the control plots. The visual impression of the sites confirmed this (see Fig. 2).

Our exemplary measurements of the maize plants indicate possible confirmation of this visual impression. The maize plants in Chitedze in the control plot were on average 92.5 cm high (n = 2, 100 cm and 85 cm of height), in the plot including gliricidia the plants reached 101.7 cm (n = 3, sd = 27.2 cm). In Makoka, the maize-only plants were 65.7 cm high (n = 10, sd = 37.5 cm), the plants in the plots with gliricidia reached an average height of 132.8 cm (n = 12, sd = 36.5 cm).



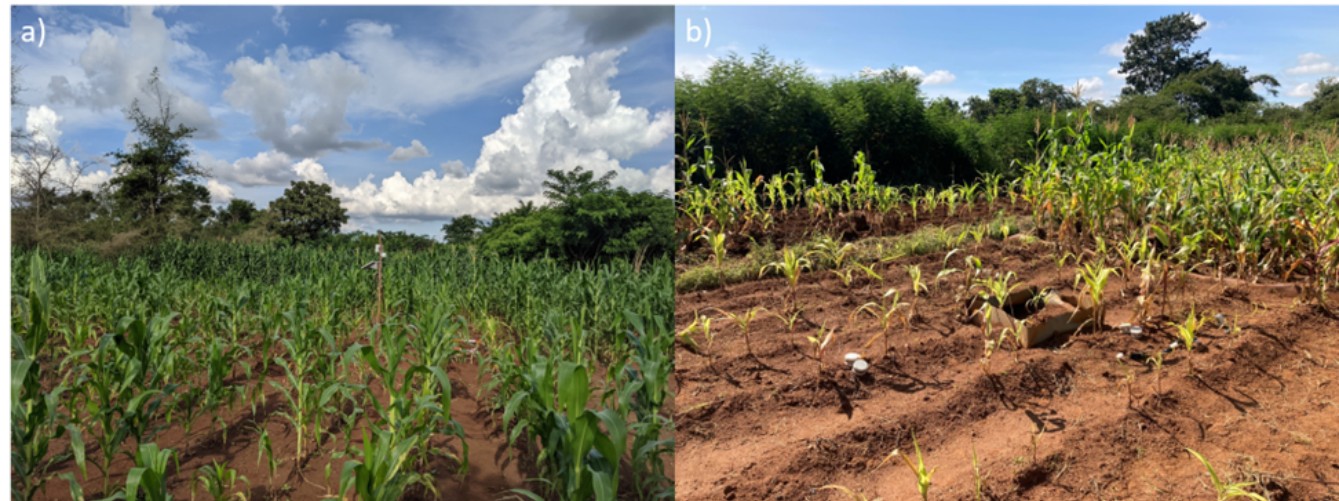

**Figure 2.** Photograph of the maize plants in maize-only control plots compared to the plots including gliricidia. a) Chitedze site: in the foreground is the control plot, in the background behind the meteo station is the gliricidia plot. b) Makoka site: In the foreground/to the left the control, to the right is the gliricidia plot.

**Table 2.** Comparison of C content in topsoil samples (0 – 20 cm) in Makoka and Chitedze at different periods after initiation of the AFSs.

| Location - Year (AFS age) | Control C content [g C kg$^{-1}$] | Gliricidia C content [g C kg$^{-1}$] | Difference C content [g C kg$^{-1}$] |
|---|---|---|---|
| Makoka - 1992 (0) [a] | 8.8 | | |
| Makoka - 2001 (9) [a] | 6.6 | 10.9 | 4.3 |
| Makoka - 2020 (28) [b] | 7.0 | 17.3 | 10.3 |
| Makoka - 2022 (30) | 7.2 | 16.4 | 9.2 |
| Chitedze - 2019 (10) | 28.7 | 30.2 | 1.8 |

[a] Makumba et al. 2006 (data from 1992, 2001)

[b] Maier et al. 2023 (data from 2020)

## 3.2 Carbon content and stability of organic matter

Makoka is the agroforestry site which has been established for a much longer time, with 30 years experiment duration. Hence, it might make sense to consider C contents of the same treatment age as in Chitedze (approx. 10 years). Makumba et al. (2006) reported for Makoka 8.8 g C kg$^{-1}$ in the topsoil at the initiation of the project and 6.6 g C kg$^{-1}$ for the control and 10.9 g C kg$^{-1}$ for the gliricidia treatment nine years after initiation (Table 2). The treatment difference after approximately ten years was therefore with 4.4 g C kg$^{-1}$ substantially larger at Makoka than the above mentioned treatment difference of 1.5 g C kg$^{-1}$

in Chitedze.




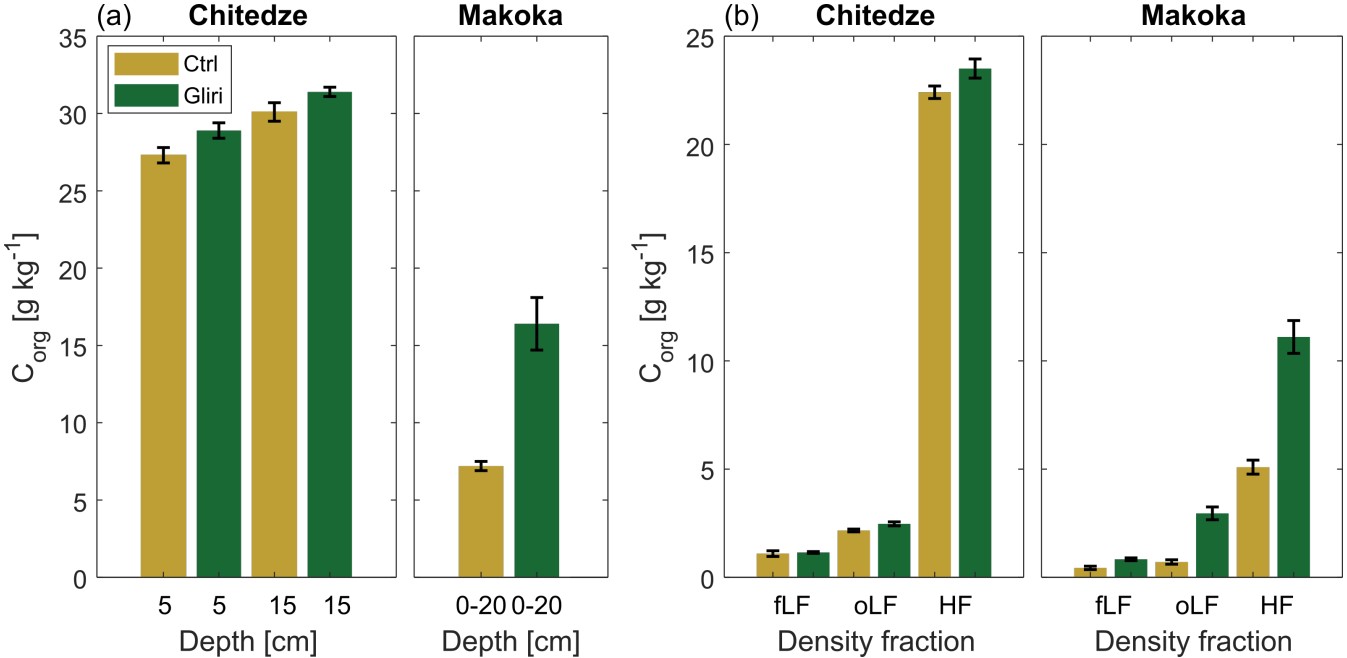

**Figure 3.** C content (a) and density fractionation of organic C (b) in the control (Ctrl = yellow) and the gliricidia treatment (Gliri = green) at the sites Chitedze and Makoka.

Soils at Chitedze showed much higher C content than soils at Makoka (Fig. 3a, Table 1). The differences between control and gliricidia were minimal at the Chitedze site, with 28.7 g kg$^{-1}$ C in the control across both depths and 30.2 g kg$^{-1}$ C in the gliricidia treatment. In Makoka, the site with generally lower C contents, the treatment effect was very pronounced with considerably higher C contents in the gliricidia plot (16.4 g kg$^{-1}$ C) compared to the control plot (7.2 g kg$^{-1}$ C).

Considering the C contents in the three different density fractions, we see that 10 years of agroforestry treatment at the Chitedze site did not lead to higher C content in any specific fraction (Fig. 3b). In Makoka, the strong increase in C contents after 29 years of agroforestry mainly occur in the aggregate-protected oLF (4-fold increase) and in the HF (2-fold increase) (Fig. 3b). A comparison of the C contents in the oLF fraction with the water-dispersible clay amounts in the 10 samples in Makoka (Table 1) showed a negative relation (Fig. 4).

Analysis of the soil density fractions revealed distinct variations in carbon and total nitrogen accumulation across fractions (Fig. 3, Table A2). In both treatments, the occluded light fraction (oLF) displayed the widest C/N ratio, followed by the free light fraction (fLF), with the heavy fraction (HF) showing the narrowest ratio.

**3.3 Soil physical and hydraulic characteristics**

Bulk density was determined at both sites as part of the soil characterisation in the small cylinders and as well as part of
the hydrological characterisation in the large cylinders. The soils in Chitedze had generally lower bulk density compared to



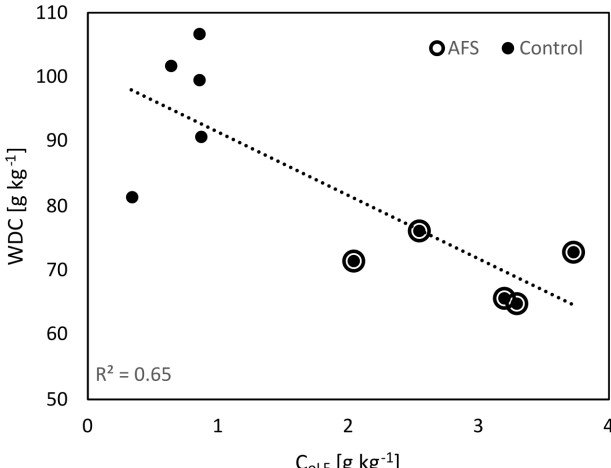

**Figure 4.** Relation between the aggregate-protected C fraction (oLF-C) and water-dispersable clay (WDC) contents at the site Makoka.

Makoka, however the variability in the data due to the small sample size is very high (Fig. 5). At both sites, bulk density clearly increased with depth. The treatment effect was more ambiguous. In Chitedze, there was no apparent difference between treatments, whereas in Makoka values of bulk density seemed to be slightly lower in the gliricidia treatment compared to the control. However, the variability again masked the difference.

Porosity was slightly higher at the Chitedze site compared to Makoka (Fig. 5). Differences with depth were mainly apparent as decreases in porosity from topsoil to subsoil. The treatment effect was not clearly visible in the comparisons. At Chitedze, values between the gliricidia plots and the controls were similar. At Makoka, the shifts were also not distinguishable from the error margin, with the possible exception of a slight increase in the subsoil porosity from control to gliricidia (Fig. 5). The treatment differences (per depth) for the small cylinders were tested with the Mann-Whitney-U test as the sample size was

sufficient and showed to be not significant.

Saturated hydraulic conductivity in the topsoil matches as expected with regard to the differences in C content and porosity, showing no difference between the control and gliricidia treatment in Chitedze. For Makoka, we found higher Ksat values in the gliricidia treatment compared to the control, as we would have expected due to also higher C contents and porosities. The subsoil painted a similar picture for both, Chitedze and Makoka. However, difference in Ksat between the treatments were

within the error bars. In all cases Ksat decreased with depth.

### 3.4    Water retention curves

Figure 6 compares the laboratory water retention curves obtained from the big soil samples to retention curves derived from the soil moisture and matric potential monitoring in the field.



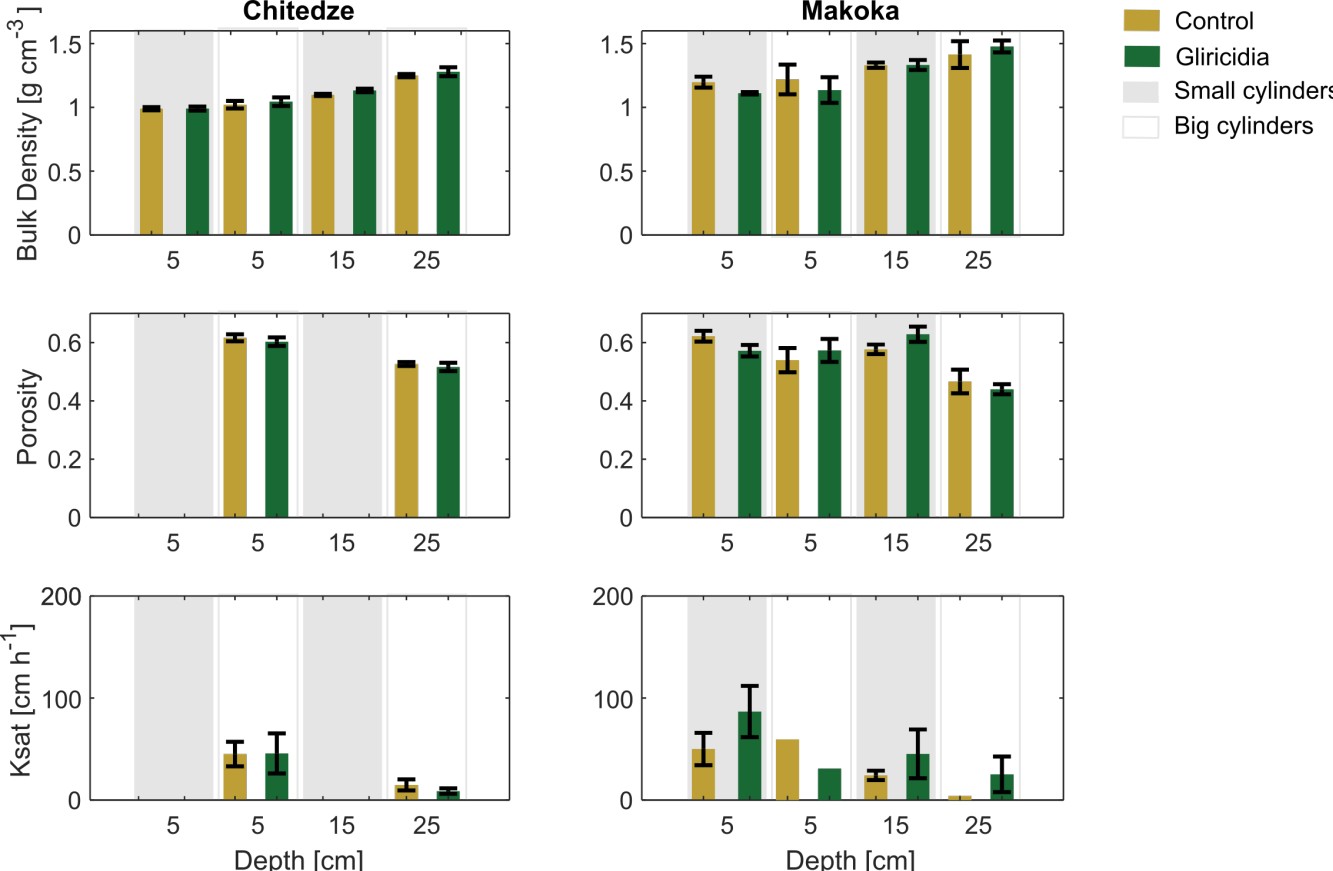

**Figure 5.** Bulk density, porosity and Ksat in 5 and 15 cm (small cylinders) and 5 and 25 cm (big cylinders) depth. For Chitedze n=10 and Makoka n=5 per treatment and depths for the small cylinders and n=3 for big cylinders. Detailed numbers can be found in the appendix (Table A1).

There is a clearly distinguishable difference between laboratory topsoil and subsoil water retention curves (Fig. 6, a and b).
The topsoil samples have always had greater porosity and flatter curves in the more humid range. This was more pronounced in Chitedze than in Makoka. Porosity values showed a similarly clear pattern: Porosities in Chitedze were overall higher than in Makoka (see section 3.3 and Fig. 4). The differences between treatments need to be considered separately for the two locations. In Chitedze, the control SWRCs were flatter in the topsoil and slightly steeper in the subsoil. Additionally, porosities were about 10 % higher in the gliricidia treatment compared to the control. In Makoka, however, there were no distinguishable
differences between in the treatments.

The comparison of laboratory (Fig. 6, lines) and field (Fig. 6 a,b, dots) retention curves, yielded an apparent "offset" around pF 0.8 − 1 at the wet end of the field retention curves. There was generally a clear difference between laboratory and field data, and the SWRC barely overlapped. The field SWRC were overall steeper and of rather similar shapes. The distance between the



**Figure 6.** Soil water retention curves at Chitedze (left) and Makoka (right). Each line curve is derived from a sample analysed in the laboratory (L) and dotted curves from field measurements (F). The solid lines represent topsoil (5 cm depth) samples (numbers in legend stand for the depth) and dashed lines for samples at 25 cm depth. The colours indicate the treatment with yellow being the control (C) and green the gliricidia (G). The grey box highlights the range of plant-available water (PAW), i.e. from pF = 1.8 (field capacity) to pF = 4.2 (permanent wilting point). The bottom panels show the calculated PAW for the different treatments and depths for the two sites.





field SWRC was, however, stronger than for the laboratory SWRC. The largest contrast was between the curves of the different

depths, SWRC of different treatments were rather overlapping. Both laboratory and field SWRC were steeper in the subsoil
compared to the topsoil.

Plant-available water (PAW) is indicated by the grey box in Figure 6 (c and d) and two interesting patterns can be observed
in the laboratory data. In Makoka, PAW in the control had slightly higher values compared to the AFS treatment, however,
generally the differences are not very pronounced. Chitedze's PAW differed between treatments with higher values in the

gliricidia.

In the field data, it is harder to identify clear patterns as the soils did not dry out until the wilting point. Nevertheless, due to
the steep slopes, it can be assumed that PAW is much smaller than the ranges observed in the laboratory.

### 3.5   Monitoring of soil moisture and matric potential

At both sites, water content and matric potential were monitored. We first consider the time series of water content.

First of all, it must be mentioned that in the control plot in Chitedze, only one of the four sensors operated (Fig 7c). Therefore,
only data from the topsoil sensor are available. In general, soil water content time series showed behaviour as expected. The
water content showed increases during rainfall events and continuous drying afterwards until the next rain event occurred.
Topsoil sensors recorded driest values and the deepest sensor the wettest values.

At a first glance, there was no clear difference in overall water content between neither the locations nor the treatments.

However, the available topsoil sensors in Chitedze as well as Makoka's soil water content measurements showed dampened,
smoothed and slower responses after rainfall in the gliricidia intercropping plot throughout all depths compared to the control.
The drying process after a rain event seemed stronger in the gliricidia compared to the control. Additionally, the drying was
more pronounced i.e. steeper in Makoka compared to Chitedze.

Further, we investigated the matric potential time series monitored at two depths in both locations and both treatments.

In a similar manner as the soil water content, the matric potential time series followed the precipitation patterns (Fig. 8). The
lower sensors at Chitedze responded almost concurrently with the top sensors at the onset of a rain event but clearly lagged
behind in the drying process. The lower sensor in the Makoka control plot (Fig. 8d) reported during the last month some
erroneous looking data potentially reaching a maximum measurement value after already being quite a bit out of its accurate
measurement range.

Nevertheless, one clear pattern observed in the treatments is that the subsoil sensors dried out stronger throughout the obser-
vation period compared to the topsoil sensors, and the control's subsoil drying exceeded the one of the gliricidia. Overall, it
appeared that more water remained available in the gliricidia treatments (pF 2-3) compared to the control (pF 3-4). Only the
sensor in the Makoka control plot (Fig. 8d) reached the PWP.

### 3.6   Analysis of rain events

Analysis of rain events, specifically, the soil's response can be insightful to understand water dynamics. In total, we identified
four rain events (> 2 mm and 6 h of inter-event period) in Chitedze and ten events in Makoka. Rainfall events in Chitedze





**Figure 7.** 15-min volumetric water content (WC) observations at both locations for both treatments respectively; colour code indicates measurement depth. Missing time series for the Chitedze maize treatment.

ranged from 3 mm to 69.4 mm, lasting between 7 h to 11 h. The Makoka events ranged from 3.8 mm to 146.4 mm and their durations from 1 h to 44 h.

To explore the soil's response to the rain events we calculated the soil water storage changes during and after an event. Exact
dates, durations and amounts are summarized in Table A3.

We show four (of five, the missing one showed only minor response in the soil water content) observed events for Chitedze in Figure 9 and point out the limitation that only the top sensor in the maize control functioned properly. In the smaller rainfall events, the topsoil sensor in the control appeared to have responded faster and sometimes also stronger compared to the sensor in gliricidia (Fig. 9, c and d), where the response was delayed or less intense. In the events with more precipitation (Fig. 9, a
and b), the water slowly percolated downwards in the gliricidia plot as demonstrated by the sequential storage increases with







**Figure 8.** 15-min matric potential observations, given as tension in pF, at both locations for both treatments, colour code indicates measurement depth. Dashed line indicates the PWP at pF = 4.2.

increasing depth. Naturally, the amplitudes slightly decreased with depth, reflecting partial storage of infiltration in the layers above. It appears that in both events (a) and (b), the top sensors reached field capacity, however, not full saturation (Fig. 6a). In Makoka (Fig. 10), fortunately, all eight sensors were functioning. Therefore, we could assess soil water storage changes in both treatments for the whole 80 cm. The largest event in the measurement period had a cumulative precipitation of 146 mm

(Fig. 10a). For this event, only a small fraction of the precipitation percolated into the soil. The largest difference appeared to be the response of the deepest sensor in the gliricidia, which was quite strong and lasted over 24 h. During the other events, which were comparable in size to the ones in Chitedze, we also observed differences between treatments. The sensors in the gliricidia plots at greater depths reacted simultaneously to the topsoil sensor. Further, the deeper gliricidia sensors showed stronger reactions than the ones in the control plot. In the control plots, the topsoil sensor showed strongest reaction and also





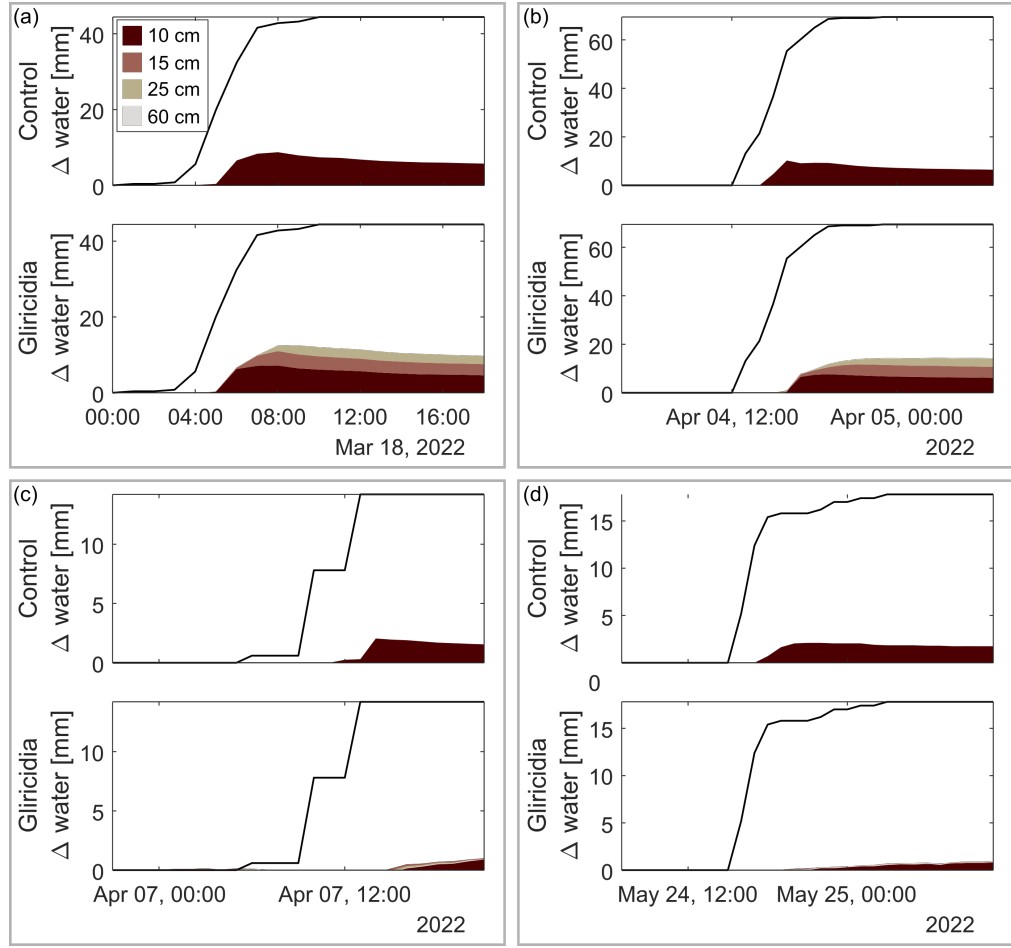

**Figure 9.** Cumulative precipitation (line) of four different precipitation events (E1-E4 from Fig. 7) and the changes in soil water storage at different depths estimated from the soil moisture sensors and separated by treatment in Chitedze.

stronger than in the gliricidia plots. The sensors in the control again responded with a stronger delay, which increased with depth. The lowest sensor reacted only occasionally to a precipitation event in both treatments.

## 4 Discussion

### 4.1 Agroforestry treatment increases C contents, relative effect is stronger in soils with generally lower C content

Two main C content patterns were observed in this study: 1) Overall the C content was higher in Chitedze (topsoil average:
29.5 g C kg$^{-1}$) compared to Makoka (topsoil average: 11.8 g C kg$^{-1}$) and 2) C content differences due to treatment effects were larger in Makoka (topsoil: 9.2 g C kg$^{-1}$) than in Chitedze (topsoil: 1.5 g C kg$^{-1}$) (Table 2). The latter providing some



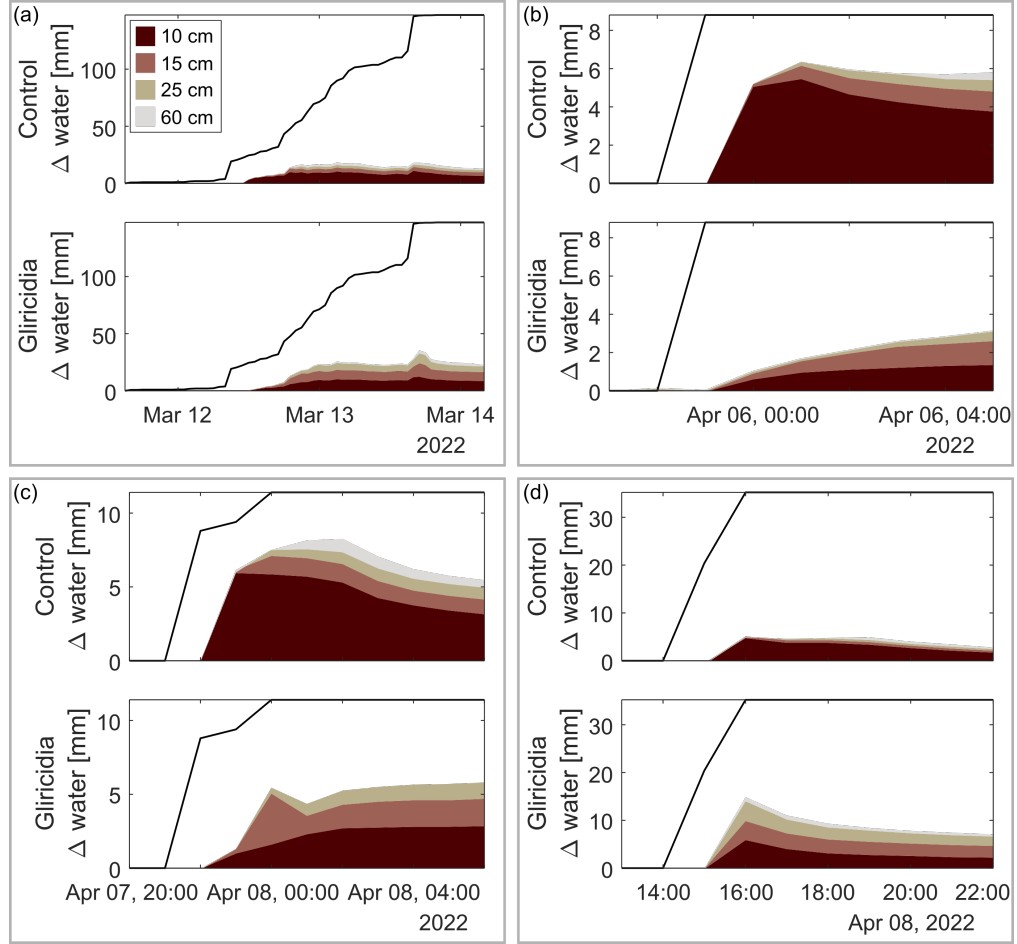

**Figure 10.** Cumulative precipitation (line) of four different precipitation events (E5-E8 in Fig. 6) and the changes in soil water storage at different depths estimated from the soil moisture sensors and separated by treatment in Makoka.

more evidence that AFS can increase soil C, supporting already existing literature (Nabuurs et al., 2007, 2022; Ramachandran Nair et al., 2009).

The treatment difference was after ten years substantially larger at Makoka than in Chitedze, which indicates that soils with
low C content store additional C more effectively as they tend to be farther from reaching their theoretical C saturation point (Maier et al., 2023). On the contrary, soils with already high C saturation, as it is the case at Chitedze, are less efficient at storing extra C and are therefore less responsive to C-input measures (Stewart et al., 2008). Also, Iwasaki et al. (2017) found a negative relationship between initial soil organic C content and the rate of soil organic C differences in Japan. Hanegraaf et al. (2009) observed highest organic C accumulation in Dutch grasslands and maize fields with lowest initial soil organic matter
content. However, they also pointed out initial soil organic matter content alone cannot explain the diverse SOM trends gener-





ally observed in their study. In soils and soil horizons with varying textures and mineral compositions, different stabilization mechanisms are dominant (Lützow et al., 2006).

## 4.2 Carbon content and pedogenic oxides affect hydraulic characteristics

### 4.2.1 Stability and aggregate formation

The soil structure plays a critical role in agricultural productivity by influencing various soil functions. One of the primary indicators of soil structure status is aggregate stability (Six et al., 2002). For Alfisols, which correspond to Luvisols and Lixisols in the US soil taxonomy, soil organic matter is the dominant factor influencing aggregation (Bronick and Lal, 2005). This finding aligns with the results of Atsivor et al. (2001), who reported that increases in soil C content can enhance soil stability through improved aggregation, particularly under sustainable agricultural practices compared to conventional farming. The stability of
aggregates depends not only on the amount of soil C but also on how C is bound within the soil matrix. WDC is an important parameter, often used as an indicator of both soil aggregate stability and erodibility (Paradelo et al., 2013).

Our results at the Makoka site indicate that the gliricidia treatment was associated with lower WDC, which suggests the formation of more stable aggregates compared to maize monocropping. Gliricidia intercropping enhanced C storage, particularly in the more stable oLF- and HF-fractions indicating free mineral surfaces left to bind and stabilize biomass C (Castellano et al.,
2015). The fLF-fraction, representing less stable C pools, remained relatively constant across treatments reflecting generally high C mineralization and C-turnover rates of rapidly decomposing N-rich legume biomass in tropical climate. The high and very high levels of pedogenic oxides (Al, Fe) at Makoka and Chitedze respectively also likely contribute to soil organic matter stabilization (Barthès et al., 2008), adding to the potential structural benefits typically attributed to increased SOM in AFS treatments.

Previous studies, such as Maier et al. (2023), observed a significant increase of more stable C fractions where C is bound to minerals and stabilized within soil aggregates. This suggests that gliricidia derived C is incorporated into soil aggregates, thereby contributing to a more stable soil structure. This structural enhancement could theoretically impact bulk density and Ksat by increasing porosity within the soil matrix. Literature supports this relationship between soil organic carbon and soil structure, indicating that soil organic carbon plays a pivotal role in promoting aggregation, stability and influencing pore size
distribution, which in turn affects water retention and water movement (Bagnall et al., 2022; Bronick and Lal, 2005; Nimmo and Akstin, 1988; Pachepsky and Rawls, 2003; Williams et al., 1983).

The potential effects of treatments on bulk density and Ksat, both of which are closely related to soil porosity, are evaluated below. Soil structure is intricately linked to the distribution of pore sizes, which, in turn, affects water movement and retention (e.g. Nimmo and Akstin (1988)). The potential changes in BD and Ksat could, therefore, provide further insight into the
mechanistic links between C sequestration, soil structure, and water dynamics under different management practices.





### 4.2.2 Comparison of soil physical characteristics and soil water dynamics

Differences in porosity, bulk density and Ksat were not directly relatable to C content in a straight forward manner. On the one hand, the expected differences in porosity were not observed, as the variations fell within the error margins. In Chitedze, a slightly higher porosity than in Makoka corresponded to overall lower bulk density. This aligns with the general under-

standing that OM and pedogenic oxides contribute to soil structure and hence affect porosity and bulk density. However, no treatment-related variations were observed. In contrast, Makoka showed measurable differences in both soil C and WDC with the treatments, which were accompanied by corresponding variations in bulk density and also Ksat.

The literature provides contradicting evidence regarding the relationship between organic matter and Ksat. Lado et al. (2004) showed that Ksat can increase in high-OM soils due to improved aggregate stability and reduced slaking. Similarly, Fu et al.

(2015) reported that high SOC content is often accompanied by increased Ksat, especially in soils with high biological activity and porosity. In Makoka, where differences in OM and soil structure were more pronounced between treatments, variations in Ksat were observed, suggesting that soil texture and mineral composition modulated the effect of OM on hydraulic conductivity.

In contrast, Nemes et al. (2005) found indications of a negative correlation between OM and Ksat. They suggest that while

OM can increase porosity, it also retains water, thereby reducing the amount of water available for free flow. This results in a complex interaction where OM enhances hydraulic conductivity by improving soil structure but simultaneously restricts water movement by retaining moisture. Our findings align with this complex relationship, as higher C content in Chitedze's soils did not result in significant Ksat changes despite higher porosity compared to Makoka.

The absence of significant Ksat changes in Chitedze could be attributed to the specific interaction between soil texture and

OM quality, highlighting the need for more detailed studies on how different OM types and soil conditions affect hydraulic properties.

Water retention curves from our study showed that topsoil samples had greater porosity and flatter retention curves in the more humid range. The treatment differences appeared smaller than differences in depth, as the former curves overlap more than the latter. The porosities of the field retention curves exceed the porosity values measured in the laboratory. The sensors were

not calibrated to the field site-specific soils, which could be one of the reasons for the difference between field and laboratory retention curves. On average, both sites demonstrated more plant-available water (PAW) in the top 25 cm in the AFS treatment; however, a treatment difference was barely noticeable in Makoka and more so in Chitedze. The literature shows varied connections between these variables. For instance, Rawls et al. (2003) reported that the relationship between soil water retention and organic carbon content is strongly influenced by soil texture. Their work suggests that in sandy soils, an increase in OM

content leads to an increase in water retention, while in fine-textured soils, the effect is less pronounced or even reversed at lower organic carbon levels. High soil C as e.g. in Chitedze's topsoil may be contributing to improved PAW, as highlighted by studies that report SOC's ability to enhance water-holding capacity (Feifel et al., 2024; Lal, 2020). However, the PAW in the Chitedze's AFS treatment exceeded Makoka's PAW. The vertical distribution of soil C also plays a critical role in the soil water balance, as indicated by Feifel et al. (2024), who found that shallow C-rich soil layers increase evaporation, while deeper





incorporation improves water availability for crops. The contrasting findings between Chitedze and Makoka emphasize the importance of site-specific factors in determining soil hydraulic properties. While OM generally improves porosity and water retention, its effect on Ksat and PAW is highly dependent on the interaction with soil texture, structure, and C distribution.

### 4.3   Changes in hydraulic characteristics are visible in water dynamics

In this section we discuss if the differences in hydraulic characterstics are also visible in the time series measurements of
soil water content, matric potential, and more specifically in their response to and after precipitation events. Based on the observed soil hydraulic properties, i.e. similar Ksat values, we expected similar infiltration responses to rainfall events at both locations. We also expected to see faster and more infiltration visible in stronger AFS sensor responses as a consequence of the treatment differences. However, we observed both, differences between sites and between treatments. Yet, comparisons were not straightforward though; between treatments in Chitedze because only the top sensor was available for the control plot, and
between the sites because the events differed in intensity and rainfall amount. Two events were relatively similar at both sites with precipitations of 12-15 and 35-40 mm. It appeared that at Chitedze the sensors reacted later after the onset of precipitation than in Makoka, where reactions happened rather immediate. Treatment differences were easier to observe. At Chitedze, the amounts of water reaching the soil differed with stronger reactions in the control plot. Especially during smaller events, the sensors in the gliricidia plots did not show a strong reaction. In addition, the sensors in Makoka's control plot measured greater
changes in water content, except for the biggest event with over 100 mm of rainfall.

One possible explanation is interception. Depending on the season, maize and gliricidia leaves cover more ground in the AFS plot. Additionally, the maize crop was larger in the gliricidia compared to the control. Larger interception capacities in the gliricidia plot potentially prevented rainfall to reach the soil and to be detected by the sensors. This could also be a reason for the different reaction times between the two sites. The maize plants were generally larger and healthier in Chitedze than in
Makoka. Another treatment difference observed in Chitedze's AFS plot was that the deeper sensors reacted with more delay to the rainfall than the top one, indicating "slow" percolation rather than macropore infiltration. This delayed response can be attributed to the fact that the gliricidia plot in Chitedze contained more clay and less sand than the control plot, which tends to reduce the speed of water movement. The lower Ksat further indicated slower water movement through the soil. In contrast, Makoka's gliricidia plot displayed an almost immediate response in the deeper sensors, which reacted nearly simultaneously
with the onset of rainfall. This rapid response suggests faster infiltration. The gliricidia plot in Makoka contained more sand, which increases soil drainage and promotes rapid water movement, and less clay, reducing water retention and allowing quicker infiltration. The soil had a higher Ksat value than the control (difference on average 123 mm h$^{-1}$) permitting water to move quickly through the soil profile. Additionally, the higher C content and greater PAW allowed more water to be stored. Similarly, Chirwa et al. (2003) found higher infiltration rates in their AFS treatment as compared to monocropping. The rapid sensor
response in the Makoka AFS might be facilitated by macropores, possibly from root structures or more biological activity in the AFS treatment compared to the control. Graham and Lin (2011) classified flow regimes based on the sequence of soil moisture sensors responses across a soil profile. Preferential or macropore flow referred to the events where deeper sensors reacted faster than shallower sensors. The deeper sensors in Makoka's control plot showed a delayed response compared to





the top sensors. This delay may be due to the absence of macropores and was potentially enhanced by the control's higher clay content relative to the gliricidia, which slowed down water infiltration and percolation. Furthermore, the lower Ksat in the top layer restricted water flow, causing moisture to move more gradually through the soil profile. One aspect to be mentioned regarding the soil water potential is the flattening of the curve during a precipitation event. The wetting does not continue until matric potential is nearly zero, but rather flattens around approximately pF=1. One reason could be that the soil has a lower (drier) air-entry potential and therefore limits water fluxes into the ceramic disk of the sensor, whose air-entry potential is at -5 kPa.

Another reason for not closing the water balance, in particular during the strong precipitation events, is that the water infiltrates at the same rate as it is redistributed into the deeper soil, hence, storage change remains stable. The high rainfall intensity of up to 30 mm during a 15 min measurement interval (120 mm h-1), does not exceed Ksat but reach close to unsaturated conductivity values (Fig. A1). The soil reaches an infiltration saturation state, therefore water remains ponding at the surface due to the absence of topographic gradients. The ponding water continues infiltrating at the rate of redistribution. The soil water measurements do not reach saturation, however, the relatively slow drying when rainfall ceased supports this. Furthermore, the water tables may slowly rise upwards along the flanks of the ridges (Fig. 2) increasing the surface area where infiltration can occur.

After rainfall events, we expected slightly slower soil drying and less water loss in Makoka due to smaller porosity, especially in the gliricidia plot, and smaller maize plants leading to less ET. In both locations, the drying effect was stronger in the gliricidia compared to the control plots, with site Makoka experiencing overall the strongest drying. This difference can likely be attributed to the higher sand content in the gliricidia plot (particularly in Makoka) enhancing drainage and accelerating the drying process. Interestingly, in both sites, the deeper soil layers in the control dried out faster and more intensely than in the gliricidia plots. This could indicate that root water uptake from maize plants was greater at depth in the control plots, possibly due to the plants accessing water reserves more effectively in response to limited surface moisture. Additionally, the plants were less protected or could grow different root structures without competition to the gliricidia roots. Matric potential measurements also indicated that more water remained available in the gliricidia treatments compared to the control plots. This suggests that the gliricidia, potentially due to higher C content, retained more water overall. It implies a generally greater storage capacity for water in the gliricidia plots, enabling them to hold moisture more effectively than the control plots. These findings are in line with reports by Chirwa et al. (2007) in AFS of gliricidia and pigeon pea in southern Malawi. They also found that there was enough water stored in the soil so that both plant species had sufficient water available.

## 4.4 Implications of introducing AFS for nutrient and water availability

In the following, we point out further aspects of introducing trees into monoculture crop fields regarding nutrients and water availability.

In terms of maize growth, the height of maize plants serves as an important measure of both yield potential and canopy interception, with taller plants indicating better growth conditions. Maize plants were generally taller and healthier in Chitedze than in Makoka, and more vital in the gliricidia compared to the control treatment. The improved growth in Chitedze may be





attributed to more favourable conditions and a higher nitrogen availability in the gliricidia treatment as observed in both the results and supporting literature on rooting depth and nutrient cycling.

Since CEC is mainly found in clay minerals and soil organic matter, it is an important factor for soil fertility and can potentially be linked to soil texture and organic matter (Beedy et al., 2010). Gaiser et al. (2012) showed that legume-derived organo-mineral compounds enhance soil CEC in a tropical Acrisol. Corresponding with consistent C contents at Chitedze, also CEC had no response on gliricidia residue input. At Makoka though we found significantly higher CEC in the gliricidia treatment corresponding to Beedy et al. (2010), who found a significant increase in soil CEC associated with gliricidia, suggesting a key

role of these trees in maintaining CEC in agricultural fields. This would furthermore explain our strong differences in maize plant size and vitality in the legume intercropping treatment compared to sole maize in Makoka.

Furthermore, we did not find any indications for severe water competition between the maize plants and the gliricidia, which was also confirmed by Makumba et al. (2009) for a gliricidia-maize intercropping system. The matric potential remained below the critical PWP at both sites under gliricidia treatment, indicating no occurrence of water stress for the plants. Chirwa et al.

(2007) found in Makoka that soil water content was generally lower in the gliricidia system at the beginning of the cropping season (end of dry season), indicating that the trees depleted soil water during the dry period. However, they also point out that rainfall exceeded potential evaporation during cropping season. If the gliricidia is pruned during crop growth, they fall into a dormant phase and do not compete for resources with the crop.

## 5 Conclusions

Introducing trees into agricultural systems in the form of AFSs may influence carbon, nutrient and water cycling in the crops. In our example of long-term experiments of maize and gliricidia intercropping we saw a clear treatment effect on soil nutrient and C contents as well as on soil structure.

We found no simple relationships between bulk density or porosity and hydrologically relevant characteristics such as hydraulic conductivity and retention properties. While there was an influence of C contents and stability on Ksat, the differences between

sites and treatments did not consistently reflect differences in bulk density and porosity. Consequently, this means that it is not straightforward to deduce changes in water fluxes from soil physical characteristics alone; the respective soil moisture and matric potential dynamics need to be measured as well for a reliable analysis of e.g. potential water availability to the crops.

In the context of climate change, we saw that some of the envisioned challenges for agriculture in Southern Africa can be alleviated with the adoption of AFSs. In the system we studied, a more stable soil structure as a consequence of the sustainably

supplied organic matter would potentially be less susceptible to soil erosion, and the better nutrient availability could sustain higher and more stable yields. A sensible combination of trees and crops, such as in the studied gliricidia with distinctly different growth habits, minimised competition for water. At the same time, the input of organic matter and changes in soil structure can buffer dry spells by protecting against fast soil desiccation and supporting the rewetting as a consequence of the next rainfall. With climate change induced shifts in precipitation patterns, this can lead to a more sustainable use of the

available water and add to the benefits of AFSs.



**Appendix A**

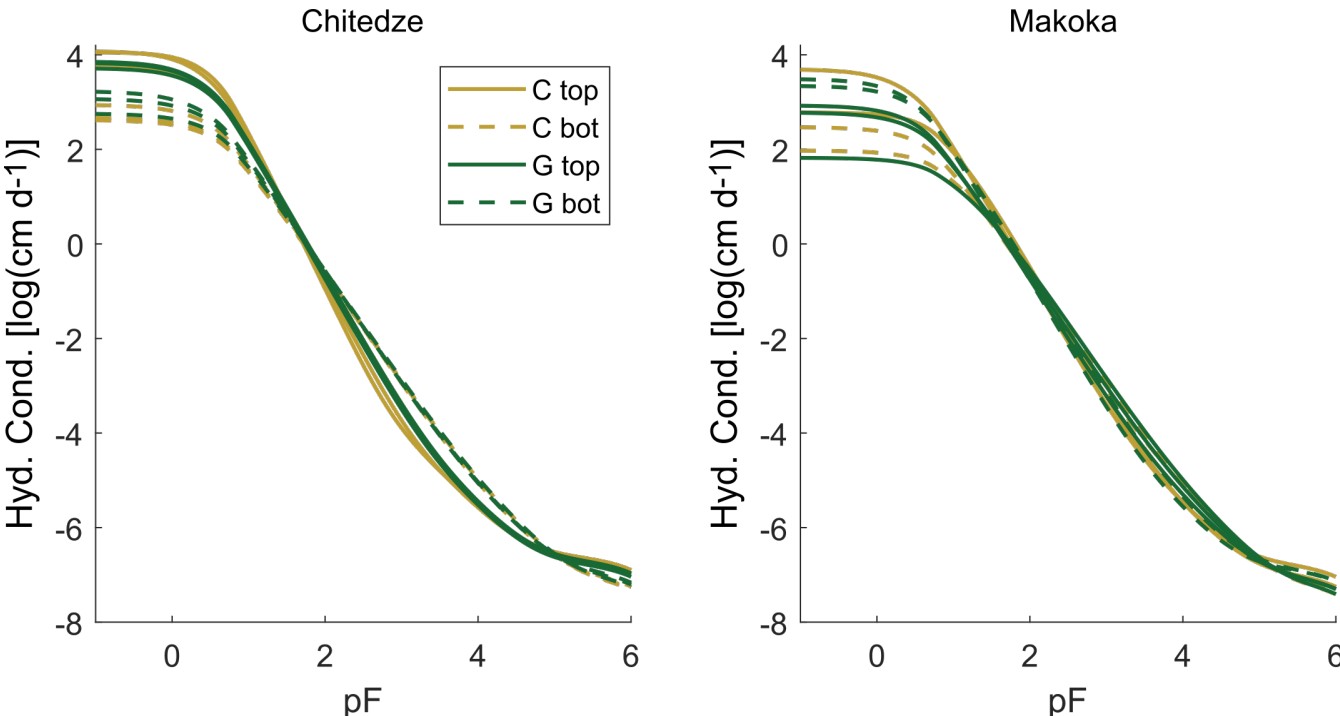

**Figure A1.** Unsaturated hydraulic conductivity derived from water retention parameters and saturated hydraulic conductivity measured in the laboratory in the control (C = yellow) and the gliricidia treatment (G = green) at the sites Chitedze and Makoka (bot = bottom).



**Table A1.** Additional soil characteristics for both sites, separated according to site, treatment and sampling depth. Abbreviations are: PAW – plant-available water; Ksat – saturated hydraulic conductivity. The values in parentheses are standard errors of the mean. The sampling year and size for the small cylinders are 2019 and 10 for Chitedze; and 2022 and 5 for Makoka. Sampling year and size of the big cylinders are 2022 and 3.

| Site | Chitedze | | | | Makoka | | | |
|---|---|---|---|---|---|---|---|---|
| Treatment | Control | | Gliricidia | | Control | | Gliricidia | |
| Depth (cm) | 5 | 15 | 5 | 15 | 5 | 15 | 5 | 15 |
| Bulk density (g cm$^{-3}$) small cylind. | 0.991 | 1.098 | 0.992 | 1.134 | 1.198 | 1.331 | 1.111 | 1.333 |
| | (0.013) | (0.009) | (0.016) | (0.014) | (0.043) | (0.021) | (0.010) | (0.039) |
| Bulk density (g cm$^{-3}$) big cylind. | 1.022 | 1.250 | 1.045 | 1.280 | 1.219 | 1.414 | 1.136 | 1.478 |
| | (0.030) | (0.013) | (0.034) | (0.035) | (0.117) | (0.105) | (0.100) | (0.047) |
| Porosity () small cylind. | | | | | 0.62 | 0.58 | 0.57 | 0.63 |
| | | | | | (0.02) | (0.02) | (0.02) | (0.03) |
| Porosity () * big cylinder | 0.62 | 0.53 | 0.60 | 0.52 | 0.54 | 0.47 | 0.57 | 0.44 |
| | (0.01) | (0.01) | (0.01) | (0.01) | (0.04) | (0.04) | (0.04) | (0.02) |
| Ksat (mm h$^{-1}$) small cylind. | | | | | 502 | 244 | 869 | 454 |
| | | | | | (158) | (46) | (252) | (239) |
| Ksat (mm h$^{-1}$) ** big cylinder | 452 | 149 | 458 | 89 | 595 | 42 | 310 | 254 |
| | (120) | (54) | (196) | (27) | (13) | (282) | (73) | (175) |
| PAW (vol. WC) * ** | 18.8 | 19.0 | 20.9 | 21.0 | 19.8 | 18.0 | 19.2 | 19.0 |
| | (1.7) | (0.6) | (0.7) | (0.5) | (2.0) | (0.6) | (1.9) | (0.6) |



**Table A2.** Density fraction of C, N and C/N values into free light faction (fLF), the occulded light fraction (oLF) and heavy fraction (HF).

| Density fraction | Chitedze | | | Makoka | | |
|---|---|---|---|---|---|---|
| | C (g kg$^{-1}$) | N (g kg$^{-1}$) | C/N | C (g kg$^{-1}$) | N (g kg$^{-1}$) | C/N |
| oLF Control | 2.17 | 0.10 | 22.53 | 0.71 | 0.03 | 23.63 |
| fLF Control | 1.10 | 0.07 | 16.59 | 0.44 | 0.02 | 18.43 |
| HF Control | 22.14 | 1.42 | 15.82 | 5.09 | 0.57 | 8.98 |
| oLF Gliri | 2.47 | 0.12 | 20.56 | 2.96 | 0.13 | 23.34 |
| fLF Gliri | 1.15 | 0.07 | 15.48 | 0.83 | 0.06 | 14.96 |
| HF Gliri | 23.51 | 1.54 | 15.27 | 11.10 | 1.14 | 9.76 |



**Table A3.** Summary of rain events measured in Chitedze and Makoka (more than 2 mm of rain, framed by periods without precipitation of minimum 6 hours).

| Event ID | Event No. (fig. 7) | Site | Amount (mm) | Start date | End date | Duration (h) |
|---|---|---|---|---|---|---|
| C1 | E1 | C | 44.4 | 18.03.2022 01:00 | 18.03.2022 10:00 | 9 |
| C2 | E2 | C | 69.4 | 04.04.2022 13:00 | 04.04.2022 23:00 | 10 |
| C3 | E3 | C | 14.2 | 07.04.2022 06:00 | 07.04.2022 13:00 | 7 |
| C4 | E4 | C | 3 | 18.05.2022 15:00 | 19.05.2022 01:00 | 10 |
| C5 | | C | 17.8 | 24.05.2022 16:00 | 25.05.2022 03:00 | 11 |
| M1 | | M | 15.4 | 11.03.2022 11:00 | 11.03.2022 18:00 | 7 |
| M2 | E5 | M | 146.4 | 12.03.2022 00:00 | 13.03.2022 20:00 | 44 |
| M3 | | M | 3.8 | 19.03.2022 06:00 | 19.03.2022 09:00 | 3 |
| M4 | | M | 8.2 | 31.03.2022 21:00 | 01.04.2022 02:00 | 5 |
| M5 | | M | 4.6 | 04.04.2022 13:00 | 04.04.2022 13:00 | 0 |
| M6 | E6 | M | 8.8 | 05.04.2022 23:00 | 05.04.2022 23:00 | 0 |
| M7 | E7 | M | 11.4 | 07.04.2022 22:00 | 08.04.2022 00:00 | 2 |
| M8 | E8 | M | 35.2 | 08.04.2022 15:00 | 08.04.2022 16:00 | 1 |
| M9 | | M | 4.2 | 12.04.2022 17:00 | 13.04.2022 01:00 | 8 |
| M10 | | M | 11 | 19.04.2022 15:00 | 19.04.2022 16:00 | 1 |



*Data availability.* The datasets that form the basis for the presented analyses are freely available and accompanied by a technical description of the individual tables from the online repository GFZ Data Services (Hoffmeister et al., 2025).

*Author contributions.* SH, SKH and RM developed the concept of the study and designed and conducted the sampling. SH, SKH and RM
collected, curated and analysed the data. SH and SKH prepared the manuscript with contributions from all the co-authors.

*Competing interests.* No competing interests are present.

*Acknowledgements.* This study would not have been possible without the logistical and field support by the colleagues at World Agroforestry (ICRAF). Konisaga Mwafongo and Charles Banda were essential for the successful field work, and Betserai I. Nyoka and the ICRAF team were always supportive with all administrative issues and background information. We are also very appreciative of the staff at the two
involved laboratories who analysed the samples for us.





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
