# Peer review of "Coupling of soil carbon and water cycles in two agroforestry systems in Malawi"

_EGUsphere, 2025_

## Referee Comment (RC2)

[referee-annotated manuscript omitted]

---

## Author Comment (AC1)

**Response to RC1**

First of all, we would like to thank the anonymous referee for their detailed and generous feedback. We put a lot of effort into preparing a concise and comprehensible manuscript and thank for the appreciation of that.

We agree that a longer monitoring of soil moisture to cover a full hydrological year and growing cycle of the maize plants would have been desirable, however, it was unfortunately not possible due to limited resources.

Thank you for pointing out that the calculations of changes in soil water storage change were not sufficiently clearly described. Soil water storage changes were calculated for each soil water content time series separately. Soil water storage is the soil water content multiplied by the sensors' measurement increment (0.05). Storage change is then derived by subtracting the soil water storage of one point in time by the previous one to get the difference between two consecutive values. The value of 0.18 was given erroneously and referring to a different sensor. We are happy to add these details to the manuscript and perhaps an equation on that matter to clarify the method.

One of the most critical points in the referee's comments is related to figure 6 on soil water retention curves, which we want to address with a bit more detail. The VGM parameters are included in the associated data publication and were not additionally presented in the manuscript as it is already very rich in data. We agree that the theta_s values, especially of the three Chitedze topsoil samples, are very high and that we did not discuss this aspect sufficiently. The VGM model reached its limitations regarding the parameter estimation and therefore "fixed" th_s and alpha in some cases to constant values (1 and 0.5, respectively), leading to the high values of theta_s as depicted in figure 6. The uncertainty bands associated with these values are extremely high (allowing a range of values of 0.28 to 1.0 for theta_s within the 2.5 and 97.5 % interval, example values from one of the Chitedze topsoil samples). As suggested by the referee, one option is setting th_s ourselves based on the small cylinder estimations but this is only possible for Makoka as porosity was, unfortunately, not measured in Chitedze's cylinder samples. The porosity values derived from the big cylinders (table A1) for both sites were estimated assuming a soil density of 2.65 g cm$^{-3}$ and were not directly measured. Another option to omit this problem would be to use the PDI model instead. This model can fit the water retention observations much better due to the separation into capillary and non-capillary flow. We decided to use VGM previously due to its wider spread in the community. We suggest using the PDI soil hydraulic model instead of the VGM and will revise the subsections accordingly.

We also welcome the referee's check on details resulting in the list of specific including technical errors that we will all gladly eliminate during the revision process. We will provide short answers to that list below (blue: original comment from the referee, black: our response).

Line 188: Last sentence of section 2.3. repeats sentence in lines 173-175.
Thank you! We will remove the repetition.

Line 211: From where does the 0.18 m come ("sensor depth increment")? It is not align with the depths indicated in line 199. Can you provide a formula for the calculation of change in water storage?
Yes, we will do that as suggested above.

Line 228: The ratio of Fe_d/Al_d was lower(!) in Chitedze
Thank you! We will adapt that.

Table 1: no texture data is given for Makoka. Why? You refer to it in lines 216-218.
Thanks a lot for pointing it out. This mistake probably occurred when formatting the table. We will include all data.

Line 236: Here you mention a C/N ratio for the intercrop of 15.4. However, in table 1 a value of 11.3 is given. Which value is correct?
Thank you. We will use the correct value of 11.3 in the text.

Table 2: Difference C content at Chitedze: 1.5 (=30.2-28.7) (instead of 1.8).
Thanks! We will correct that.

Line 249: 4.4 gC kg$^{-1}$ or 4.3 gC kg$^{-1}$ (table 2)?
Thanks, we will correct it to 4.3 in the text as correctly stated in table 2.

Line 252: unit g C kg$^{-1}$ ff.
OK. We will look once again through the manuscript to ensure that the same unit description is used.

Lines 267-8: lower bulk density in gliricidia sites only at 5 cm, but reverse at 15 cm!
We will clarify this in the text and adjust the sentence.

LineS 277-278: In my opinion, this statement is misleading as it contradicts the data from the large cylinders. These show a decrease in Ksat! Indeed, data from the small cylinders show an increase, but that goes along with a decrease in porosity. Here it is certainly better to rely more on the results from the large cylinders, which are probably more reliable.
We understand the contradiction. However, we believe that the data from the small cylinders are more robust as the number of samples taken was substantially larger than for the big cylinders. We can add this point to the manuscript. Also, we will remove the remark on differences in porosities and add instead that the difference were marginal and therefore cannot be considered interpretable. That way hopefully reducing confusion and uncertainty.

Figure 6 gives me some puzzles. I am irritated by the presented retention curves for Chitedze measured in the laboratory, as they show saturated water contents of 0.75-

0.95 m³/m³ (assuming that x-axis indicates volumteric soil water content as indicated in the label). How can this be? Values of this magnitude are not realistic. They also contradict the results from soil cylinders. In which pF-range were the retention curves actually recorded using the Hyprop? What are the values of the van Genuchten parameters (eq. 1) derived from Hyprop measurements? When fitting the retention curve to Hyprop data, it would probably have been more expedient to fix theta_sat to the value measured by soil cylinders. Since the PAW values are directly derived from these retention curves, I also cannot fully trust them.

We have made some suggestions for how to tackle these concerns above and are happy to adapt the manuscript accordingly.

Figure 8, figure caption: E5-E8 in Fig. 7 instead of Fig. 6 Thanks for pointing this out. We will change it!

A few abbreviations are not defined before first use ("BD", "OM") We will check again and include the explanation of the abbreviations that are missing.

Table A1: re-arrange rows and indicate differences in sampling depth for small cylinders (5, 15) and larger cylinders (5, 25) .

Table A3: When comparing figs. 7 & 9 with tabele A3, I think event E4 corresponds to C5 with 17.8 mm (instead of C4 with 3 mm only) Yes, we agree. It should be E4. We will correct that.

---

## Author Comment (AC2)

**Response to RC2**

We would like to thank the referee for their thorough and detailed feedback. We appreciate the large effort put into reading the manuscript and making valuable suggestions. We will go through all comments in the following.

**Legend for colour code**

Black – referee comments

Grey – additional referee comments from PDF

Green – our response

**1. Summary**

The manuscript titled "**Coupling of Soil Carbon and Water Cycles in Two Agroforestry Systems in Malawi**" addresses an important topic relevant to sustainable agriculture and climate adaptation in Southern Africa. It explores the coupling of soil carbon and water cycles within maize-Gliricidia-based agroforestry systems in Malawi, drawing on experimental data from well-established long-term research sites. Conducting research on such long-term experiments provides valuable insights into the sustained impacts of agroforestry management on soil health and water dynamics over time.

Given the increasing challenges posed by climate change in Southern Africa, understanding how agroforestry influences both carbon sequestration and water availability is critical for developing sustainable land-use practices. This study's focus on soil carbon content, physical psroperties, and hydrological responses under Gliricidia-maize intercropping and maize monocropping provides important information for enhancing agroecosystem resilience.

**2. General**

While the topic is relevant and timely, the manuscript would benefit from clearer explanations of data analysis, consistent use of terminology, and more thorough contextualization of the results. Detailed feedback is provided below.

A2.1. Thank you for the detailed feedback on these aspects. We agree that the manuscript will benefit from adapting these points. We will give detailed answers to each point below.

There is also mixing of sections, with literature frequently cited in the Results section, reflecting an organizational problem. The manuscript should be structured according to the journal's guidelines. Furthermore, some information that belongs in the Methodology section is instead presented in the Results section; see, for example, lines 265 and 304-305.

Line 245: What is this? you are mixing the section! you can discuss this in the discussion section.

A2.2. Thank you for pointing this out. We will address some specific literature references in the remaining response where the referee highlighted specific points and made suggestions for improvement. Furthermore, we will go once more through the manuscript and check for misplaced literature or rather mixing of sections.

Line 248: I usually see the flow where the results are presented first, followed by the tables or figures below, but this seems like a different style.

A2.3. This flow arises automatically using the LaTex template provided by the journal. The figures/tables are placed nearby their first mentioning but also fitting to the overall format e.g. to avoid half empty pages. Anyhow, formatting will be taken care of by the journal's technical team.

Line 249: Mostly the carbon content is presented in percentage unit why you prefer to present is g C kg-1?

A2.4. Thank you for pointing this out. As $g\,kg^{-1}$ is a SI unit and is used in most publications in terms of soil organic carbon we decided to handle it the same way.

**3. „Controlled" AFS**

Consistency in Terminology: The abstract refers to "controlled setup," which may be misleading since the experiments are conducted under field conditions. Clarifying this will avoid confusion. see answer A3.1

Line 103. Sometimes you use the wrong words. For example, calling it a "controlled agroforestry experiment" is odd for an experiment conducted under field conditions. see answer A3.1

Line 103. Misleading information is included, such as referring to the study as a controlled agroforestry experiment. A controlled experiment has a very different meaning, whereas the agroforestry experiment in question was conducted under field conditions. see answer A3.1

A3.1. Thank you for pointing this out. We will clarify this and replace the terminology of "controlled setup/experiment" with "field conditions/experiments" in all instances to avoid confusion.

**4. "Undisturbed" soil samples and sampling depths**

Line 125: Would the ridges in the maize plot intercropped with gliricidia trees affect soil sampling for bulk density and other parameters that require undisturbed soil samples for analysis? see answer A4.1

Line 142: There is a lack of information regarding the soil sampling system and sample collection. Since there are ridges inside the plots, how did you manage to take samples

from both disturbed and undisturbed soil depths (e.g., see line 170) in both treatments at the two sites? Furthermore, some studies report that in tree-based agroforestry systems and annual crop production areas, the 31-90 cm soil layer typically represents the depth where leached nutrients accumulate and where root growth contributes organic matter and nutrients. Do you think sampling only the 0-10 cm and 10-20 cm layers is sufficient to represent the actual soil conditions at the site? I also have doubts about the depth ranges (e.g., 0-10 and 10-20 cm); I personally suggest using 0-10 cm and 11-20 cm ranges. see answer A4.1

Line 144: Why do you want to change the name when the actual conditions on the ground are different? see answer A4.1

Lines 141-147, line 170, and line 199 show inconsistency in sample collection for physicochemical and hydrological data collection. Sometimes you use depth intervals of 0-10 cm, 11-20 cm, other times 0-20 cm, or specific depths like 5 cm, 7 cm, 10 cm, 15 cm, 25 cm, and 60 cm, which makes it very difficult to follow the overall research flow. There is consistent inconsistency in sampling and sample depth selection, making it hard to interpret the results. This variability in sampling could itself introduce additional variation beyond the treatment effects (control and Gliricidia agroforestry plots). see answer A4.1

Line 170: ?????? Why are your depth ranges changing from time to time? Why did you choose 5 cm and 25 cm instead of the above mentioned intervals such as 0–10 cm, 11–20 cm, or 15 cm? see answer A4.1

Line 197: What depths exactly? Please provide the list of depth ranges (e.g., 0–10, 11–15, 16–25, and 26–60 cm). I don't understand why you randomly selected those depth ranges could you please explain your reasoning? see answer A4.1

Line 200: There is a consistent inconsistency in sampling and sample depth selection, making it difficult to interpret the results. This variability in sampling could itself be an additional source of variation, beyond the treatment effects (control and Gliricidia plots). see answer A4.1
Pleases correct the ranges. 0-10, 11-15, 16-25, 26-60 cm or other system see answer A4.1

As described in the Methods section, the presented result data show a lot of inconsistency in depth ranges for soil physicochemical and hydrological characteristics, which makes it quite confusing and hard to understand the overall flow of the results. see answer A4.1

Figure 3: This should be clearly defined! what do mean 5 cm and 15 cm, you have to show the ranges (0-5 cm and 0-15/6-15 cm). see answer A4.1

**A4.1. Reply to "undisturbed" soil samples and sampling depths**

**Undisturbed soil samples**

Thank you for inquiring more thorough explanation of our sampling methods. To clarify: We took samples from within the maize ridges at both treatments (control and intercropping). Yes, we agree, by piling up the ridges and planting the maize plants within them each growing season, the soil is being disturbed. Two thoughts on that: 1) This treatment and the resulting disturbance of the soil represents the typical management in this region and is therefore interesting to study as it controls soil water flow; 2) the soil in between the ridges is where practitioners frequently walk to work the fields whereby also compacting it. In managed/agricultural fields, it is basically impossible to find undisturbed/natural soil without human impact. The aim of our sampling was to get an assessment of the soil conditions on such fields. The samples are undisturbed in the sense of being taken directly from the experimental site, with as little disturbance as possible. With that we captured the structural state of the soil at time of sampling. Doing so, we achieved volume related soil sampling.   This information will be added in the manuscript.

**Sampling depths**

Thank you for letting us know that the depths were not defined well enough and this impaired the flow of reading. We will add a table to the manuscript explaining which samples were taken where and what analyses were conducted on them, similar to the summary table in the data publication. We will additionally add a sketch to the appendix illustrating the sampling design. We will further change the names of the samples to ranges based on sampling depth and cylinder height throughout the whole manuscript.
We chose the depths based on the soil depths of interest. First of all, we were interested in the upper soil characteristics. That's why we took samples in the topsoil near the surface. The lower depth is supposed to represent subsoil with its different characteristics.

**"Table XX**. Overview of the samples we took and the performed analyses. Abbreviations of headers: Loc = location, No. = Number of samples per depth, SM = sampling method, Lab = laboratory, where samples were analysed/processed, BD = bulk density, P = porosity, C, N = carbon and nitrogen content, DF = density fractions/fractionation, PO = pedogenic oxides. T = texture, WDC = water-dispersible clay, Ksat = saturated hydraulic conductivity, WR = water retention curve, P_WR = porosity from water retention curve. The depth is given in cm.

| Loc | Year | Depth | No. | SM | Lab | BD | P | C,N | CEC | DF | PO | T | WDC | Ksat | WR | P_WR |
|---|---|---|---|---|---|---|---|---|---|---|---|---|---|---|---|---|
| C | 2019 | 0-10, 11-20 | 10 | A | UF | | - | X | X | | X | | - | - | | |
| C | 2019 | 0-10, 11-20 | 10 | CS | UF | X | - | | | | | | - | - | | |
| C | 2019 | 0-20 | 10 | A | UF | | - | | | X | | X | - | - | | |
| C | 2022 | 5-10,25-30 | 3 | CB | KIT | X | | | | | | | | X | X | X |
| M | 2022 | 5-9,15-19 | 5 | CS | UF | X | X | | | | | | | X | | |
| M | 2021 | 0-20 | 5 | A | UF | | | X | X | X | X | X | X | | | |
| M | 2022 | 5-9,25-29 | 3 | CB | KIT | X | | | | | | | | X | X | X |

M = Makoka

C = Chitedze

A= auger samples (disturbed)

CS = cylinder samples (undisturbed), small cylinders

CB = cylinder samples (undisturbed), big cylinders

UF = Chair of Soil Ecology, Institute of Forest Sciences, Faculty of Environment and Natural Resources, University of Freiburg, Germany

KIT = Chair of Hydrology, Institute for Water and Environment, Karlsruhe Institute of Technology (KIT), Germany"

[Figure]

**5. Data from different sources, locations, times**

In line 134. Previous data were included in this article, but details about the data and the source link (local or international repository) were not provided. I personally recommend presenting this information in the appendix section of the article. Additionally, you should mention it in the introduction for example, by stating that previous work has addressed certain aspects, but further analysis is still needed due to a lack of [specific information or data] or inconsistencies in the results (short- or long-term).Furthermore, It is a fact that both pedological and hydrological characteristics vary with time and season, so you should explain why you chose to include data from 2019 and 2021. see answer A5.1

Why do you want to compare the different soil and hydrological characteristics between sites? It is quite clear that the soil classes at the two sites are different (i.e., Chromic Luvisols vs. Ferric Lixisols). Differences are expected due to soil formation processes, parent materials, and stages of weathering. see answer A5.1

Line147-150: The sampling times vary, with some campaigns conducted in 2021 and others in 2022, which is methodologically challenging for soils in tropical areas. Both soil and hydrological characteristics change over time, and using data from different periods could complicate the results and overall findings of this research. see answer A5.1

Why didn't you analyze all the physiochemical characteristics from the soil samples collected in 2022? Using different year data could complicate the overall result and findings. This is methodological very wrong.. see answer A5.1

Line 223: I don't understand why you are interested in comparing the pedogenic oxides between sites that already have different soil classes (Chromic Luvisols in Chitedze vs. Ferric Lixisols in Makoka). see answer A5.1

Table 1: How can you compare samples collected in 2019 in Chitedze to samples collected in 2021 in Makoka? Further, the number of samples and depth ranges are even not equal in the two sites (eg. look you comparison in line 228). In which tools and method you analyze the data? I am very much interested to know! I feel that the tables are just a part of the report with little value in helping to understand the main objective of this research and the topic in general. see answer A5.1

There is inconsistency in the number of samples used for analysis in the results presented in the tables. For example, results analyzed with 10 samples at one site are compared with results from only 5 samples at the other site. For example in Table 1, the soil characteristics from 2019 (Chitedze) with a larger sample number were compared with those from the other site (Makoka) in 2021, which had only 5 samples. This is methodologically incorrect, if you don't you proper analysis tools for such type of data. Additionally, there is no information on the number of replications for each treatment at the two sites. see answer A5.1

Line 251. You are comparing the C content between sites. In my opinion, these two locations are incomparable due to differences in sampling and methodology. It would be better to focus on the treatment effects at each site separately. see answer A5.1

Line 349: Still hard to make such type of conclusion within the context of differences in lot of situations between the two sites. see answer A5.1

Line 431: Your focus is not on the location; rather, your discussion should focus on comparing the treatments at the two sites separately. see answer A5.1

Line 435: So how can you make a comparison between sites when they differ in both rainfall intensity and amount? see answer A5.1

A5.1. Answer to data from different sources, locations, times
We understand from your questions that it may appear confusing to readers which samples, methods and analyses were used. We hope that by making the changes as explained in answer A4.1, the sampling design and methods will be clarified. The existing data were collected during a different campaign. Nevertheless, they were extremely valuable to us to bridge the gap between soil properties and soil water dynamics. Given the resources at hand, we designed the field campaign for collecting further data to extend the existing data sets on soil properties and also sample hydrological characteristics. This is how we ended up with data from

different years and with different sample sizes. We agree, that certain properties change over time. However, the time interval of 1 or 2 years is probably much less relevant than the 10 and 30 years of differences in treatment.
The aim was to compare the different treatments, not the sites. The information on the two sites are being used to explain differences in the treatment effects. The importance of pedogenic oxides in connection with C stabilization is mentioned in line 378 (Barthès et al., 2008). When focusing on C input measures, C storage and stabilization, the different contents in pedogenic oxides are indeed relevant as well as soil textural differences and other soil properties influencing C storage process. We will streamline the methods and results to make sure that the treatment effect is the main focus.

**6. Concerns regarding hydrological measurements**

Line 173: The majority of the measurements, such as hydraulic conductivity (Ksat), were conducted in the laboratory, despite the availability of accepted field measurement methods. Why choose to measure Ksat using laboratory instruments in Germany when field measurements using a double-ring infiltrometer or Guelph permeameter are easier and more cost-effective than transporting soil samples to a distant location? see answer A6.1

Line 194-195: The falling-head method is generally recommended for fine-textured soils due to its sensitivity to low permeability. Could you clarify why this method was selected, especially if the soil in your study includes coarser textures? For example, Guelph permeameter and double-ring infiltrometer.

A6.1. Thank you for this comment. Yes, there are a number of measurement methods to derive information on soil water flow, which all have their advantages and disadvantages.
We agree that using infiltrometers are valuable site assessment strategies, however, they also have to deal with assumptions about the flow field propagation into the subsurface, which are often not really valid (Angulo-Jaramillo et al., 2000). Moreover, infiltration capacity (from field measurements) and hydraulic conductivity (from soil sample analyses) are not exactly identical. Furthermore, the usability of these rather large devices within the maize plants and ridges is questionable.
In our laboratory, we have a lot of experience with the methods chosen for this study and can acquire high precision. We can use the very same sample to receive several values/information (e.g. saturated hydraulic conductivity, soil water retention curves, soil texture) as opposed to one value for each separate measurement from the suggested methods, which allowed more information gain for the same amount of resources used.

Angulo-Jaramillo, R., Vandervaere, J.-P., Roulier, S., Thony, J.-L., Gaudet, J.-P.,

Vauclin, M., 2000. Field measurement of soil surface hydraulic properties by disc and ring infiltrometers. Soil Tillage Res. 55, 1–29. https://doi.org/10.1016/S0167-1987(00)00098-2

Line 195: I feel it would be better to refer to scientific articles that have used the same instrument for rainfall measurement.

A6.2. This is a common instrument and standard method for rainfall measurements, so we believe there is no need to cite literature using the device.

Line 204: Despite the actual rainy season being from November to April, why were you interested in collecting some hydrological data (e.g., water flux data such as soil moisture and matric potential) from March to May 2022? Did you also consider calculating antecedent soil moisture content?

A6.3. For the context of our study, we were specifically interested in the transition from the wet to drier state because this is when competition between plant species is especially critical. Interesting aspects include most of all the drying or wetting patterns and not so much the only wet (always sufficient water) or only dry (constant water deficit) state. Therefore, we decided to start measuring in the wet season and continued into dry season to see how the soils react to the subsequently drier conditions.
We refer to answer A13.25 for our response on antecedent soil moisture.

Line 210: The soil water storage change was calculated based on the difference between two successive soil water content measurements. At what time intervals did you conduct the soil moisture and matric potential measurements, and why did you select those intervals?

A6.4. Soil moisture and matric potential measurements were conducted at 15-minute intervals, which is mentioned several times in the manuscript. However, it was not included in the method section. We will add this relevant missing information. The interval was chosen as a balance between storage and power resources on the one hand and information gain on the other hand. This interval proved to be suitable for these assessments in previous studies.

**7. Statistical analyses**

In general, there is no information on data analysis, such as the type of software used to analyze the soil and hydrological characteristics, or how the mean values of these characteristics were compared between treatments at the two sites. It is also difficult to conduct ANOVA with only two treatments. I suggest performing a two-factor factorial analysis for each site, introducing treatment and depth as factors. Additionally, if interested, you could consider treatment and site as two factors to analyze their interaction effect. see answer A7.1

L223: What do mean higher without any statistical analysis? With what criteria you reported the value as higher? see answer A7.1

Line 224: I am very worried about how you will convince the readers without showing any statistical analysis like a t-test or ANOVA. It is not even possible to conduct ANOVA (Analysis of Variance) because it is designed to compare the means of three or more groups to determine if at least one group differs significantly from the others. Further, how do you compare two sites with different depth ranges? see answer A7.1

Line 229: What type of statistical tool did you use, and how did you compare the means? There is nothing mentioned about this in the methodology section, yet the statistical analysis results appear like magic. see answer A7.1

The entire Results section lacks statistical t-test analyses with significant p-values and letters indicating differences between treatments (e.g., a, b, c in mean values in the tables or figure bars). In general, the soil physicochemical and hydrological characteristics are presented simply as high or low without any statistical comparison or significance values. see answer A7.1

Line 235: How can you come to this withour conducting statstical analysis? see answer A7.1

Line 249: Prove it with t-test! see answer A7.1

Line 265: sometimes you compare soil characteristics between sites, but if interested, you could instead perform a two-factor ANOVA to examine the interaction effect of site and treatment. As I understand it, you want to evaluate the effect of Maize-Gliricidia based agroforestry compared to maize monocropping; therefore, it would be better to compare these results separately for each site.

Line 265: Why do you always want to show this? You could instead perform a two-factor ANOVA to examine the interaction effect of site and treatment. As I understand it, you want to evaluate the effect of gliricidia-maize as an agroforestry system compared to maize monocropping; therefore, compare these results for the two sites separately. see answer A7.1

Line 274: The treatment differences (per depth) for the small cylinders were tested using the Mann-Whitney U test, as the sample size was sufficient and results showed no significant difference. However, this is not the correct place to explain the analysis method; such details should be included in the Methodology section. see answer A7.1

Line 286: The same concern! see answer A7.1

Figure 5: Show the statistical analysis results with p-values and 'a' and 'b' symbols on top of the bar in Figure 5. see answer A7.1

Line 373: It is not correct for this study case because we don't know wether the treatments have statstical differences in cabon between the treatments or not? see answer A7.1

Line 390. The fact is that there is no statistical difference in bulk density and Ksat between the two treatments under different management practices. How do you support your argument in light of this? see answer A7.1

Line 393: On what type of comparison did you base this conclusion, given the considerable biophysical variability and the differing histories of establishment between the two sites? see answer A7.1

Line 397: We did not see any statistical evidence showing a variation in BD and Ksat between the treatments (e.g., Figure 5) see answer A7.1

4413: "The treatment differences appeared smaller than differences in depth" this means that depth is another important factor, or that depth mattered more than the treatment in explaining the differences. I suggest conducting a two-factor analysis to assess the interaction effect between treatments and depth. see answer A7.1

A7.1. We agree that we often compared the mean values directly without testing for significant differences. In some instance, we used the Wilcoxon-Mann-Whitney (Mann-Whitney-U) test to check for significant differences, but missed adding these analyses to the methodology part of the manuscript. We will include this into the method section with the according references.
We will add Mann-Whitney-U tests of remaining parameters (where possible) to support the comparisons and add the missing information. As suggested and taken back by the referee, we agree that ANOVA is not the right tool to use in our case due to the small size of our data sets.

Wilcoxon, F., 1945. Individual Comparisons by Ranking Methods. Biometrics Bull. 1, 80. https://doi.org/10.2307/3001968

Mann, H.B., Whitney, D.R., 1947. On a Test of Whether one of Two Random Variables is Stochastically Larger than the Other. Ann. Math. Stat. 18, 50–60. https://doi.org/10.1214/aoms/1177730491

**8. Literature**

Line 30: I couldn't find this reference on the web.

A8.1. Thank you for pointing this out. We checked the reference again and the report is not available on the website anymore. Therefore, we will remove it and instead refer to an IPCC special report:

IPCC, 2019a: Climate Change and Land: an IPCC special report on climate

change, desertification, land degradation, sustainable land management, food security, and greenhouse gas fluxes in terrestrial ecosystems[Skea, J., E. Calvo Buendia, V. Masson-Delmotte, H. O. Pörtner, D. C. Roberts, P. Zhai, R. Slade, S. Connors, R. van Diemen, M. Ferrat, E. Haughey, S. Luz, S. Neogi, M. Pathak, J. Petzold, J. Portugal Pereira, P. Vyas, E. Huntley, K. Kissick, M. Belkacemi and J. Malley (eds.)]. Shukla, P. R., In press pp. Available at: https://www.ipcc.ch/srccl-report-download-page/  (accessed 25/10/2020).

Line 40: Add these supporting articles such as: Iwasaki et al., 2009; Breedy et al., 2010; Alamu et al., 2023.

Here is Alamu et al., 2023 full citation: Alamu, Emmanuel Oladeji, et al. "Gliricidia sepium (Jacq.) walp applications for enhancing soil fertility and crop nutritional qualities: a review." Forests 14.3 (2023): 635.

A8.2.  Thank you for these valuable suggestions. We will gladly add the suggested literature to the manuscript and the reference list.

Line 72: What do you mean by "farmers might be concerned"? Is there any evidence or findings that show how farmers perceive the impact of agroforestry systems (AFS) on soil evaporation?
What do you mean tree component?
add reference at the end of this text: or competition for water in the root zone (....)

A8.3.  Thank you for pointing out our unfortunate formulation. There are a few studies that looked into agroforestry adaptation obstacles, however, not many specifically for the tropics. Mercer (2004) summarised in a review agroforestry adoption innovations in tropical regions and conclude that most concerns stem from household decision patterns, available resources, market incentives, biophysical conditions, and the farmer's perceptions of risk and uncertainty. Valdivia et al. (2012) conducted a study in Canada where they found that next to cost factors for establishing and managing trees (machinery, personnel) and timing of investment returns also reduced crop yields due to competition of resources (light, water and nutrients) played a crucial role in the decision of establishing agroforestry systems. Reported disadvantages such as increased transpiration and competition for water in the root zone have been demonstrated in different case studies. We will reformulate and correct the sentence and its literature references: "On the other hand, competition over resources such as water may occur (Odhiambo et al., 2001; Siriri et al., 2013), which is also a perceived obstacle for the implementation of AFS (study in Canda, Valdivia et al., 2012)."

By "tree component" we are referring to trees as one of the components of an agroforestry system. We will rephrase the sentence to avoid confusion. "Farmers might be concerned to implement AFS due to a potential negative impact of

implementing trees next to or within a crop field...".

Mercer, D.E., 2004. Adoption of agroforestry innovations in the tropics: A review. Agrofor. Syst. 61–62, 311–328. https://doi.org/10.1023/B:AGFO.0000029007.85754.70

Odhiambo, H.O., Ong, C.K., Deans, J.D., Wilson, J., Khan, A.A.H., Sprent, J.I., 2001. Roots, soil water and crop yield: Tree crop interactions in a semi-arid agroforestry system in Kenya. Plant Soil 235, 221–233. https://doi.org/10.1023/A:1011959805622

Siriri, D., Wilson, J., Coe, R., Tenywa, M.M., Bekunda, M.A., Ong, C.K., Black, C.R., 2013. Trees improve water storage and reduce soil evaporation in agroforestry systems on bench terraces in SW Uganda. Agrofor. Syst. 87, 45–58. https://doi.org/10.1007/s10457-012-9520-x

Valdivia, C., Barbieri, C., Gold, M.A., 2012. Between Forestry and Farming: Policy and Environmental Implications of the Barriers to Agroforestry Adoption. Can. J. Agric. Econ. Can. d'agroeconomie 60, 155–175. https://doi.org/10.1111/j.1744-7976.2012.01248.x

Line 74: Any literature support for this statement?

A8.4. Thank you for pointing this out. For instance, Bayala et al. (2014) state in their review that a key factor deciding on the balance between advantages over disadvantages is dependent on the "species and the ecological context". Further, Sileshi et al. (2014) point out that the great potential of optimal use of agroforestry for nutrient cycling is management dependent. We will add these two references to the manuscript:
"This balance is very much dependent on the climatic conditions and the combination of the species in the AFS (e.g. Bayala et al., 2014) and can also be influenced by appropriate management (Sileshi et al., 2014)."

Bayala, J., Sanou, J., Teklehaimanot, Z., Kalinganire, A., Ouédraogo, S., 2014. Parklands for buffering climate risk and sustaining agricultural production in the Sahel of West Africa. Curr. Opin. Environ. Sustain. 6, 28–34. https://doi.org/10.1016/j.cosust.2013.10.004

Sileshi, G.W., Mafongoya, P.L., Akinnifesi, F.K., Phiri, E., Chirwa, P., Beedy, T., Makumba, W., Nyamadzawo, G., Njoloma, J., Wuta, M., Nyamugafata, P., Jiri, O., 2014. Agroforestry: Fertilizer Trees, in: Encyclopedia of Agriculture and Food Systems. Elsevier, pp. 222–234. https://doi.org/10.1016/B978-0-444-52512-3.00022-X

Line 74: As readers, we want to see a coherent, informative, and engaging introduction. Your background should briefly explain what has been done so far and clearly identify the gap in the research. Avoid assigning extra reading to the reader by referring them to other literature to understand the importance of your study. Instead, craft all the key points and present them in a clear and concise way within your introduction..

A8.5. Thank you for the reminder on how to structure an introduction. We will remove the sentence referring to Sheppard et al. (2020) to avoid confusion. This manuscript is not focusing on general advantages and disadvantages of agroforestry systems. The for the study important aspects are mentioned in the following lines (76-79).

Line 76. The findings were not fully presented while making the argument. However, they are still influenced by seasonal variation. For instance, Chirwa et al. (2007) reported that the available soil water content at the start of the cropping season was generally lower in tree-based systems, indicating that trees likely continued to extract soil moisture throughout the dry season.

A8.6. Thank you for the suggestion. We will discuss this point thoroughly in the paragraph on previous studies at the sites (see answer A8.7). We will add to the sentence at hand "...while during seasonal effects some competition may occur" to ensure it is not being simplified too much.

Line 93: In my opinion I luck details review findings of the previous research works related to the topics in Malawi!

A8.7. Thank you for pointing this out. We will gladly include a short summary of previous research conducted at the field sites in line 86. "Previous studies conducted at these research sites encompass research on maize yields and nutrient dynamics (Akinnifesi et al., 2006, Ikerra et al., 1999, Makumba et al., 2006a, Chirwa et al., 2003, Akinnifesi et al., 2007), rooting patterns (Makumba et al., 2019), carbon sequestration (Makumba et al., 2006b), soil organic matter (Beedy et al., 2010) an its stabilization (Maier et al., 2023) and soil water dynamics (Chirwa et al., 2007); and drought resilience (Kerr 2012) in maize/gliricidia intercropping systems. The gliricidia intercropping system proved to increase maize yields if managed correctly by pruning gliricidia regularly and applying prunings into the soil (Akinnifesi et al., 2006; Chirwa et al., 2003). The gliricidia intercropping had a positive effect on soil and particulate organic matter, which were directed at increasing yields and storage capacities for nutrients (Beedy et al., 2010). The addition of inorganic N and P fertilizer together with organic inputs from the gliricidia positively influences maize yield (Akinnifesi et al., 2007). Gliricidia redistributed N from the subsoil to the surface (Ikerra et al.,1999), however Makumba et al. (2006) still found a net decrease in gliricidia systems due to increased nutrient export. Maize was shown to have more roots growing within 0 -

40 cm depth than gliricidia and could therefore benefit from the nutrients of the applied gliricidia prunings in the ridges (Makumba et al., 2019). The intercropping also had beneficial effects on biomass production and carbon input as well as improved aggregate formation and storage of SOM within aggregates (Maier et al., 2023). Furthermore, the intercropping system could sequester more carbon in the soil than maize plants alone (Makumba et al., 2006b) and Chirwa et al. (2007) found that under typical rainfall conditions, water was sufficiently available for plants to grow and water use efficiency increased in the intercropping system. Kerr (2012) found that gliricidia intercropping improved maize production even under conditions of drought stress."

Ikerra, S.T., Maghembe, J.A., Smithson, P.C., Buresh, R.J., 1999. Soil nitrogen dynamics and relationships with maize yields in a gliricidia-maize intercrop in Malawi. Plant Soil 211, 155–164. https://doi.org/10.1023/A:1004636501488

Chirwa, P.W., Ong, C.K., Maghembe, J., Black, C.R., 2007. Soil water dynamics in cropping systems containing Gliricidia sepium, pigeonpea and maize in southern Malawi. Agrofor. Syst. 69, 29–43. https://doi.org/10.1007/s10457-006-9016-7

Beedy, T.L., Snapp, S.S., Akinnifesi, F.K., Sileshi, G.W., 2010. Impact of Gliricidia sepium intercropping on soil organic matter fractions in a maize-based cropping system. Agric. Ecosyst. Environ. 138, 139–146. https://doi.org/10.1016/j.agee.2010.04.008

Akinnifesi, F.K., Makumba, W., Kwesiga, F.R., 2006. Sustainable maize production using Gliricidia/Maize intercropping in Southern Malawi. Exp. Agric. 42, 441–457. https://doi.org/10.1017/S0014479706003814

Makumba, W., Janssen, B., Oenema, O., Akinnifesi, F.K., Mweta, D., Kwesiga, F., 2006. The long-term effects of a gliricidia–maize intercropping system in Southern Malawi, on gliricidia and maize yields, and soil properties. Agric. Ecosyst. Environ. 116, 85–92. https://doi.org/10.1016/j.agee.2006.03.012

Makumba, W., Akinnifesi, F.K., Janssen, B.H., 2009. Spatial rooting patterns of gliricidia, pigeon pea and maize intercrops and effect on profile soil N and P distribution in southern Malawi. African J. Agric. Res. 4, 278–288.

Makumba, W., Akinnifesi, F.K., Janssen, B., Oenema, O., 2007. Long-term impact of a gliricidia-maize intercropping system on carbon sequestration in southern Malawi. Agric. Ecosyst. Environ. 118, 237–243. https://doi.org/10.1016/j.agee.2006.05.011

Chirwa, P.W., Black, C.R., Ong, C.K., Maghembe, J.A., 2003. Tree and crop productivity in gliricidia/maize/pigeonpea cropping systems in southern Malawi. Agrofor. Syst. 59, 265–277. https://doi.org/10.1023/B:AGFO.0000005227.69260.f9

Akinnifesi, F.K., Makumba, W., Sileshi, G., Ajayi, O.C., Mweta, D., 2007. Synergistic effect of inorganic N and P fertilizers and organic inputs from Gliricidia sepium on productivity of intercropped maize in Southern Malawi. Plant Soil 294, 203–217. https://doi.org/10.1007/s11104-007-9247-z

Maier, R., Schack-Kirchner, H., Nyoka, B.I., Lang, F., 2023. Gliricidia intercropping supports soil organic matter stabilization at Makoka Research Station, Malawi. Geoderma Reg. 35, e00730. https://doi.org/10.1016/j.geodrs.2023.e00730

Kerr AC (2012). Drought resilience of maize-legume agroforestry systems in Malawi. University of California, Berkely, USA. PhD Thesis. https://escholarship.org/uc/item/7bm3k6nv

Line 164: What are the similarities and differences between this study and the one by Hoffmeister et al. (2025)? Is there any connection to this study (e.g., in terms of data, methods, etc.)? What is the purpose of mentioning it here?

A8.8. Hoffmeister et al. (2025) is a data publication of the data used for this study. In that way we make the data freely available to the public following some standard data layout and detailed explanations, to support open science and reproducible research.

Line 357: Some supporting literature is presented in the discussion without relevant information. These references are off-topic, not related to agroforestry systems or maize, and are from non-tropical climatic zones. In research focused on the tropics, your citations come from Japan and Germany. Couldn't you find any literature related to tropical regions?

A8.9. Yes, that is correct. There is research on certain agroforestry topics in the tropics but for some aspects we could not find any literature from tropical zones. Therefore, we expanded the research to outside the tropical zones. In a global analysis, Minasny et al. (2017) indicate a relationship between initial carbon stock and carbon sequestration rate. However, it is not possible to specifically pick out tropical regions, also this study is not related to agroforestry systems. We will add this reference to the manuscript and emphasizing the lack of tropical studies and therefore accurate assessment for our study.

Minasny, B., Malone, B.P., McBratney, A.B., Angers, D.A., Arrouays, D., Chambers, A., Chaplot, V., Chen, Z.-S., Cheng, K., Das, B.S., Field, D.J., Gimona, A., Hedley, C.B., Hong, S.Y., Mandal, B., Marchant, B.P., Martin, M., McConkey, B.G., Mulder,

V.L., O'Rourke, S., Richer-de-Forges, A.C., Odeh, I., Padarian, J., Paustian, K., Pan, G., Poggio, L., Savin, I., Stolbovoy, V., Stockmann, U., Sulaeman, Y., Tsui, C.-C., Vågen, T.-G., van Wesemael, B., Winowiecki, L., 2017. Soil carbon 4 per mille. Geoderma 292, 59–86. https://doi.org/10.1016/j.geoderma.2017.01.002

Line 366: Have you identified any literature on the relationship between organic matter and soil aggregation in tropical regions, particularly in Africa? You cite Bronick and Lal (2005), but the argument you present is not consistent with the WRB/FAO classification (i.e., Alfisols are distinguished from Luvisols and Lixisols).

A8.10. Thank you for this remark. We refer to a study by Ayuke et al. (2019) investigating the influence of land use practices and management on soil aggregation and SOM dynamics. They found aggregate stability indices and soil organic matter to be generally higher in the fallow compared to the arable systems. Soil organic matter build up significantly enhanced aggregate stability in their study. Six et al. (2002) found a higher aggregate stability but lower correlation of this stability with carbon contents in tropical soils compared to temperate soils.
We will remove the comparison of the different soil types and instead include the two references from tropical regions.

Ayuke, F.O., Zida, Z., Lelei, D., 2019. Effects of Soil Management on Aggregation and Organic Matter Dynamics in sub-Saharan Africa. African J. Food, Agric. Nutr. Dev. 19, 13992–14009. https://doi.org/10.18697/ajfand.84.BLFB1002

Six, J., Feller, C., Denef, K., Ogle, S.M., de Moraes, J.C., Albrecht, A., 2002. Soil organic matter, biota and aggregation in temperate and tropical soils - Effects of no-tillage. Agronomie 22, 755–775. https://doi.org/10.1051/agro:2002043

**9. Title**

It is better to modify the title because the two agroforestry systems are not actually different. Both sites use a Gliricidia sepium-based agroforestry system, so there is no need to refer to them as two separate systems.

Here is my suggested title: Coupling of Soil Carbon and Water Cycles in *Gliricidia sepium*-Based Agroforestry Systems in Malawi. Still I have question on the term "water cycles".

A9.1. Thank you for the suggestion of adapting the title and questioning its relevance. We agree that "water cycles" might sound misleading, therefore we will change the term to "soil carbon and water dynamics". Otherwise, we will leave the title as is to avoid being too specific about the type of agroforestry system and shifting the focus from the soil carbon and water dynamics to the specific agroforestry system. We will happily include *Gliricidia sepium*\* to the abstract for those interested specifically in this system to find it quickly.

**10. Abstract**

The abstract provides a clear overview of the study's objectives, methodology, and key findings related to soil carbon and water dynamics in Maize-Gliricidia-based agroforestry systems and Maize monocropping in Malawi. It effectively highlights the importance of agroforestry in addressing climate challenges and improving soil health.

However, some areas could be improved for clarity and precision:

Focus on Key Results: While the abstract summarizes results well, some sentences could be more concise. For example, the relationship between soil physical properties and water dynamics could be stated more directly.

Contextualization: The abstract would benefit from briefly mentioning the implications of the findings for sustainable agriculture or land management in Malawi.

No keywords were provided

Overall, the abstract is informative but could be polished to improve readability and impact.

A10.1. We will gladly modify the abstract to meet the referee's suggestions and concerns. Therefore, we focus on changes in the results paragraph, streamlining it and emphasizing implications of the results:

"Our results show a clear increase in C contents and stability as a result of the gliricidia impact compared to the control at the site with the generally lower baseline C contents. At this site, the treatment effect was not visible in soil physical characteristics such as porosity and bulk density, but in saturated hydraulic conductivity, which is rather a structural property. The soil water dynamics were influenced by several additional factors such as soil texture and interception. The gliricidia treatment showed greater soil water storage capacities and retained overall more water, while generally none of the plots neither control nor treatment were under severe water stress during the observation period. We also noticed a protective effect against soil drying below the topsoil potentially by more immediate/macropore infiltration into the subsoil under gliricidia.
We conclude that from a methodological point of view assessing the effects on water fluxes requires respective field measurements as they cannot be deduced from soil physical characteristics directly. Overall, the AFS treatment of adding gliricidia into maize cultivation can have a considerable effect on nutrient and water dynamics in the crop, however, this effect is also dependent on initial site conditions. A sensible AFS implementation can not only support carbon accumulation and stabilization but also increase the efficient use of available water thus supporting different aspects towards sustainable agriculture in Malawi."

Biogeoscience does not require keywords.

**11. Introduction**

I noticed some redundancy in the introduction section of the article. For example, the effects of intercropping Gliricidia trees with maize are repeatedly discussed in relation to soil structure, soil fertility, soil organic carbon (SOC) content, carbon stocks and stability, water dynamics, and maize yield. These points should be streamlined and organized clearly to avoid repetition.

A11.1. Thank you for this valuable feedback on the structure of the introduction. We will revise and streamline it to avoid redundancy (see also A11.3).

Additionally, I observed a disruption in the flow of ideas within one paragraph. You wrote, "Consequently, in this study we examine the link between carbon input and soil hydrology," which is immediately followed by, "There is still a lack of knowledge about the extent to which carbon-induced short- and long-term changes in soil structure affect water fluxes in these systems." This sequence weakens the coherence of the introduction.

A11.2. Thank you. We will rephrase the paragraph to streamline the introduction and to maintain coherence.

I recommend revising the introduction to improve clarity, coherence, and engagement. The background should succinctly summarize existing knowledge, highlight the specific knowledge gap, and effectively convey the importance of your study without redundancy. It would also be helpful to clearly and concisely differentiate your research from previous studies such as Chirwa et al. (2007), Hoffmeister et al. (2025), Kerr (2012), Kirsten et al. (2021), Maier et al. (2023), and Makumba et al. (2006, 2007).

A11.3. Thank you. We believe that this comment is partly addressed in the literature section (A8.7), where we formulate a paragraph on existing research at the sites. We will go further through the introduction to make sure we meet the referee's advice on clearly differentiating our work and motivation based on previous research. We will streamline the introduction as follows:
- paragraph on importance and advantages and disadvantages of agroforestry systems: transition effect of trees on soil carbon
- current state of research on impact of agroforestry treatment on soil physical and hydraulic properties (there are several paragraphs on this topic, which will be streamlined to achieve more coherence)
- introduction of the knowledge gap
- introduction of the study site, its relevance and previous research conducted there
- stating our research questions

Line 44: Since this is the first mention, the full scientific name should be written out, including the authority who first identified it (e.g., Gliricidia sepium (Jacq.) Steud.).

A11.4. Thank you for noticing this. We will write out the full name as suggested.

Line 58: What do you mean? is it to mean intercropping crops with legume trees as AFS has been shown to increases soil aggregation? Pleases add soil!

A11.5. Thanks for pointing that out. We will rewrite the sentence as suggested.

Line 64: You should clearly define what the tree component is?

A11.6. We don't understand this question as the tree component is not mentioned here at all. However, during our revision and streamlining process of the introduction, we will pay attention to clearly mention the tree component.

Line 72. Some of the findings are not fully reviewed or presented in the introduction section. For example, the relationship between increased SOC and soil evaporation is still debated in the literature and may depend on contextual factors such as soil texture and climate, as reported by Feifel et al. (2024).

A11.7. Thank you for pointing this out. We will add the points suggested by the referee to line 74: "This balance is very much dependent on the climatic conditions, the combination of the species in the AFS and other factors like actual SOC concentration, depth of humic layer and soil texture (Feifel et al., 2024)".

Line 83. Lack of information is provided concerning the distance between the two sites. Sites that are 316 km apart were generalized as being in the same climatic zone. Does this make sense? If so, please show us on a map, using a DEM or a table with long-term climate data from meteorological stations.

A11.8. Thank you for this comment. Unfortunately, we do not have a DEM or long-term climate data available. However, we checked with the climate classification system. According to the Köppen classification (Beck et al. 2023) are the two study sites located in two different climatic regions. Chitedze belongs to the Cwa classification, which is a temperate climate with dry winters and hot summers. Makoka belongs to the Aw category, which describes a tropical Savannah climate. We will include this information to the site description.

Beck, H.E., T.R. McVicar, N. Vergopolan, A. Berg, N.J. Lutsko, A. Dufour, Z. Zeng, X. Jiang, A.I.J.M. van Dijk, D.G. Miralles High-resolution (1 km) Köppen-Geiger maps for 1901–2099 based on constrained CMIP6 projections Scientific Data 10, 724, doi:10.1038/s41597-023–02549-6 (2023)

It is better to include the hypothesis! It would strengthen the manuscript to explicitly state the study's hypothesis (or hypotheses) in the introduction. Presenting the hypothesis provides readers with a clear understanding of the research's expected

outcomes and the rationale behind the study design. This also helps in framing the results and discussion, allowing the audience to assess whether the findings support or refute the initial assumptions.

A11.9. We agree that formulating hypotheses or research questions, which have been the basis for our study, supports the structure and understanding of the text. However, in our understanding, the formulation of research questions is more suitable in this context as little research has been conducted on specifically the influence of soil carbon structural changes on soil hydrological properties. Therefore, the formulation of evidence-based hypotheses is not possible. This is why we introduced three research questions at the end of the introduction.

**12. Method**

I feel that it is better to provide topics for each section of the methodology, such as: Study Area Description, Experimental Design, Description of Agroforestry System / Cropping System, Sampling Procedures, Measurement Techniques, Data Collection on Environmental Variables, Laboratory Analysis, Data Analysis Methods, Quality Control and Assurance, and Ethical Considerations or Permissions.

A12.1. Thank you for this suggestion. We understand that our approach might be in some cases a bit atypical by dividing subsections based on different sampling types. However, as you have noticed and correctly pointed out in several comments, we do have some inhomogeneity in our sampling methods, times and locations. That's why we decided to structure the methodology section in the way we did, so that we can guide the readers through these different approaches. We agree that this can be improved in some parts of the section but we believe structuring it solely on above mentioned subsections will create confusion as we would have to jump forth and back between different samples for each of these subsections.

We will streamline the method section to make sure that we maintain the same pattern throughout and will include a subsection on data analyses for each sampling method. We will also add a sentence to the beginning of the method section, which will explain the structure so that the reader already knows what to expect.

Our structure is the following:
2.1 Site description and management *(this basically includes the referee's points of study area description, experimental design, description of agroforestry system/cropping system without naming them specifically)*
- Chitedze
- Makoka
2.2 Soil sampling with auger and small cylinders *(sampling procedures, measurement techniques)*

- Chitedze
- Makoka
- Laboratory analyses of auger and small cylinders
- Data analyses of auger and small cylinders
2.3 Soil sampling with large cylinders *(sampling procedures, measurement techniques)*
- Chitedze
- Makoka
- Laboratory analyses of large cylinders
- Data analyses of large cylinders
2.4 Monitoring of meteorological and hydrological variables (*data collection on environmental variables*)
- Equipment and installation
- Data analyses of monitoring data

The management practices in the control (maize monocropping) were not explained at all. If it was conducted without any fertilizer, it does not reflect the reality of maize production in Malawi or other African countries. It would make more sense to compare maize production under local fertilizer rates and typical management practices. The experiment would have been improved by including three treatments: control (no fertilization), T1 (fertilizer at the recommended local rate), and Maize-Gliricidia based agroforestry.

A12.2. We shortly explain management practices at the sites in lines 121 to 126. We will reformulate the paragraph to clarify that: "In the agroforestry treatments, gliricidia was planted between alternate ridges with a spacing of 1.50 m between ridges and 0.90 m within the ridge, resulting in a planting density of about 7,400 trees per hectare, as outlined in a previous study by Makumba et al. (2006).
The management of all plots (maize control and gliricidia AFS) follows common practice of planting the maize on ridges each year before the rainy season starts. Additionally, on the gliricidia plots the trees are cut three times a year to a height of 30 cm, in October, December, and February and tree pruning (leaves and tender branches) are incorporated into the soil ridges as green manure."

Furthermore, in this study we were not interested in maize production in general or under local fertilizer rates, therefore this suggestion is not as relevant to us. We wanted to understand how the gliricidia trees impact the soil carbon and how this in turn influences soil hydraulic properties and dynamics. Therefore, looking at the control and agroforestry was the main goal, including different fertilizer rates would open up a whole new topic, which would require a new and different study.

Line 102: The map is not sufficiently explanatory, as it lacks essential elements and does not clearly represent the study area's landscape and elevation. Could you please

provide a DEM map of the study area that includes all necessary features, such as coordinates, a legend, and other relevant details? This DEM map will help to clearly highlight the climate zones within your study area.

A12.3. Thank you for this comment. As mentioned before, unfortunately, we don't have a DEM map of the study areas available. However, we will gladly expand the description of the study areas to the different climate zones. The map is merely meant as a rough source of orientation for people not familiar with Malawi, not as an additional source of climate or elevation information.

Line 105: Information that should be presented in the introduction, which could help to highlight the research gap, is misplaced here in the Methods section. For example, see line 105.

A12.4. This comment refers to text lines describing the study site, which we believe is placed correctly in the method section under site description and management. After all, our study focus is on treatment effects and not on the site comparison, which merely helps to understand why treatment effects differ between sites.

Line 115: There is a lack of consistency in reporting temperature: for the Chitedze site, the mean annual temperature is reported, whereas for the Makoka site, the mean daily temperature is given.

A12.5. Yes, we agree there is an inconsistency in reporting temperatures. Unfortunately, we could not find reliable data to harmonize to either mean annual or mean daily temperature. Therefore, we decided to report the existing published values instead of none.

Line 116: Lack of consistency in soil group classification by the IUSS Working Group: For the Chitedze site, you cited IUSS Working Group (2014) for the classification of Chromic Luvisols, while for the Makoka site, you used IUSS Working Group (2022) for Ferric Lixisols. I suggest using the latest version (e.g., IUSS Working Group, 2022) for both sites. Typically, such data are established at the country level, and you may be able to find updated classifications for both locations in the newer version.

A12.6. We agree and we will correct the inaccuracy in the manuscript by using only the IUSS Working Group WRB from 2022 for both locations. "Chitedze had less sand than Makoka and therefore qualified as clay loam and Makoka with a greater sand fraction as sandy clay loam (according to IUSS Working Group WRB (2022)".

Figure 1. Figure 1 requires changes based on the above comments.

A12.7. Thank you for your suggestions. We can unfortunately not accommodate all, mostly because we don't have a DEM available. However, we suggest adding a sketch illustrating sampling locations and depths.

Line 135: Some of the topics seem overly broad and include irrelevant information. It would be better to provide concise main topics and split the rest into subtopics. For example, having 'Soil sampling with an auger' as a main topic in the article seems quite unusual.

A12.8. We understand that this seems unusual. We addressed this point previously and suggested changes (first comment on methodology section) that will hopefully improve the structure of this section.

Line 138: There is a lack of standardized names for sampling instruments. Use 'Kopecky rings' instead of 'small and large cylinders,' as Kopecky rings are standardized tools for collecting undisturbed soil samples in the field. Since there are ridges inside the plot, how did you manage to collect undisturbed soil samples from these areas? To what extent can the samples collected from this site be considered truly undisturbed?

A12.9. Thank you for this remark. In the scientific literature circulate many terms for this specific tool: Kopecky rings, soil sample rings, soil cylinders and more. We decided on soil cylinders and introduced the two different cylinder sizes with their specific measures and volumes.
Part two was addressed above in the section on undisturbed sampling.

Line 143: Does this mean that density fractions and texture are not affected by depth? As I understand it, factors such as soil erosion and human disturbances (like tillage and ridging) vary with depth and can influence soil texture. Likewise, variations in organic matter, root activity, and microbial activity with depth can, in turn, affect soil density fractions.

A12.10. Thank you for this important question. It was not our intension to convey that there are generally no depth-related differences in soil texture or density fractions, these differences undoubtedly exist. Since the ridges have been rebuilt annually since the beginning of the experiment with the topsoil permanently mixed in the process, we decided to mix the samples for some additional analyses. As other research on SOC in Malawi refers to 0-20 cm depths, we wanted to compare the specific depths range with existing results. We will rephrase the sentence to: "For density fractionation and the analysed soil texture, the samples from two depths were combined and analysed as mixed samples referring to 0-20 cm soil depth. This way we were able to compare the results from the different sites."

Line 154: How reliable is the distilled water method for measuring water-dispersible clay (WDC), considering that some clay particles are strongly bound and will not disperse without the use of chemical dispersants such as sodium hexametaphosphate ($Na_6P_6O_{18}$)? Furthermore, the results can be influenced by soil type.

A12.11. Thank you for this comment. We will add the missing citation to the manuscript (van Reeuwijk, 2002.). The WDC method with distilled water is specifically

recommended to be used for soil characterization in the World Reference Base for Soil Resources, which we followed.

van Reeuwijk, L.P., 2002. Technical Paper 09: Procedures for Soil Analysis (6th Edition) | ISRIC, 6th ed. Internation Soil Reference and Information Centre

Line 154: You analyzed CEC but you don't analyzed $Na^+$? Please also correct the notation for $Ca^{2+}$, $Mg^{2+}$, and $K^+$.

A12.12. Thanks for pointing that out. $Na^+$-cations were also analysed though not taken into consideration. We will correct the notation for cations in the manuscript.

**13. Result**

You are only presenting the results in the Results section, but it is important to include explanations of your own reasoning, justifications, and the implications of the high and low values of the soil and hydrological characteristics between treatments.

A13.1. We did not find any clear guidelines by the journal specifying how to specifically separate results and discussion. We decided to report results without interpretation in the results section and everything that goes beyond that into the discussion. In the discussion section, we state our own reasoning and interpretation of the results and compare it and provide further background with the help of existing literature.

Line 218: This is not a discussion section where you refere a citation to support your finding. You can show the percentage of the texture class (sand, silt and clay) and percent your result.

A13.2. Thank you for pointing this out. We will remove the reference here, because we are only describing the soil texture results.

Line 220: Please at the end of the text mention where we could see this result (Table… figure …..). Further, no texture result for Makoka site in table 1.

A13.3. Thanks for mentioning this. We will include the texture data for Makoka and include in the text where mentioned data can be seen as suggested.

Line 225: What value?

A13.4. Thank you for noticing this. We are referring to the Fed values at Makoka. We will add "$Fe_d$" to the sentence.

Line 227: Compared to what you said high?

A13.5. We will reformulate the sentence. "Both sites have levels above 28 g $kg^{-1}$ $Fe_d$ and above 3 g $kg^{-1}$ $Al_d$."

Line 228: ????? Which Fed/ALd ratio is higher?

A13.6. Thanks. We agree the formulation of this sentence leads to confusion. We will update it to: "The ratio of $Fe_d/Al_d$ was in Chitedze 6.5 for the control and 6.3 including gliricidia compared to Makoka with 8.8 for the control and 9.3 including gliricidia."

Table 1. No information about the sand, silt and clay faction for Makoka site 0-20 cm depth ranges.

A13.7. Thank you for pointing this out. We do indeed have those data and used them also to conduct the study. We will of course add them to the table in the updated manuscript.

Table 1. The table captions sometimes include too much information, such as abbreviation details. They should follow the standard format of the journal.

A13.8. We understand that our table captions are sometimes very long and contain much information. However, as the journal submission guidelines state "the tables should be self-explanatory and include a concise, yet sufficiently descriptive caption", there is a need to explain abbreviations that are not common to everyone. We will try to shorten and streamline table captions to keep them as concise as possible:

"Table 1. Soil characteristics (texture, pedogenic oxides and nutrients) according to site, treatment and sampling depth. Numbers behind the site name indicate sampling year (y) and sample size (n). The values in parentheses are standard errors of the mean. For texture in Chitedze, we also show the averages of the two depths for easier comparison with the Makoka data. Abbreviations are: WDC – water-dispersible clay; $Fe_d/Al_d/Mn_d$ – dithionite-extractable Fe/Al/Mn; CEC – cation exchange capacity."

Table 1: What is this? If it is a standard deviation, you should include the ± sign.

A13.9. As explained in the table caption this is the standard error. We will include the +/- sign.

Figure 2. It is very hard to understand and visualize the differences in Figure 2a, which seems to be based on subjective judgment. In Figure 2b, at the Makoka site, the experiment is designed as a randomized complete block design. It is important to show how the other replications look so that it is easier to visualize the differences. In general, since these figures are not the major aim of this research, it would be better to include them in the supplementary section of the article. Furthermore, no standard sampling procedures was reported for this data.

A13.10. The photos are intended to exemplify visual comparison between controls and treatments. We didn't intend to provide quantitative information with this but to illustrate the differences.

Previous research in Makoka always referred to all blocks and showed that the differences between replicates did not overshadow differences between treatments. As we were interested in the treatment differences, we chose one pair of control and intercropping, to focus our resources on.

Table 2-: It would be better to move this table to the supplementary section, as it is not directly relevant to the Results section because it presents results from previous research. I am unsure of the intention behind including secondary data in the Results section, especially since soil properties (such as soil carbon) change over time. You could discuss these changes in the Discussion section while referring to the data in the supplementary materials.

A13.11. As table 2 includes also data from our research we would keep it in the result section. The data from previous studies are used to complement our data set, therefore we find it suitable in this context. We are applying a space-for-time approach for our study to monitor temporal changes of hydrological characteristics that can be attributed to the agroforestry treatment. As we have additional information from previous studies we can look at the temporal evolution at the site.

Line 248: What is the reason for this and what is the implication should be highlighted here.

A13.12. We are elaborating on this in the discussion section 4.1.

Figure 3. A beautiful figure was presented but they statistically don't show anything and seem like static, meaningless figure.

A13.13. From our perspective is this key figure of the manuscript highlighting how the treatment effect manifests differently at the two locations. On the left hand (a), we can observe an increase in soil carbon concentration in the gliricidia plots compared to the control and it also demonstrates that this treatment effect is larger in Makoka. Subplot (b) depicts the density fractions that carbon is bound to and which can be attributed to aggregate stabilization. Again, treatment differences are larger in Makoka. Therefore, we believe that this figure is a key component of the manuscript.
We will add stars for significant treatment differences to the figure to emphasize its relevance.

Line 264: You already mentioned these on the methodology section and no need to repeat it here.

A13.14. The first sentence was thought as a short introduction to the next subsection. We will remove it from the manuscript.

Figure 4: The relationship between WDC and Colf was negative, but the $R^2$ value ($R^2$ = 0.65) is positive. Additionally, the legend for AFS is not clearly identified in Figure 4.

What is the implication of this from the point of soil health?

A13.15. $R^2$ is simply the squared correlation coefficient and quantifies the explained variance. The correlation can of course be positive or negative as in this case -0.8 (antiproportional).
WDC can be used as an Aggregation/Erodibility Index. The higher the WDC-values are, the more clay particles get dispersed in water and washed away with rain and the other way around: the smaller the amount of WDC the less prone to erosion the soil is.

Line 278: Bulk density (BD), porosity, and Ksat values did not show any statistical evidence of differences between treatments at the two sites. For example, although Ksat was higher in the Gliricidia treatment than in the control, this difference was not statistically significant (see Figure 5).

Why don't you show the correlation between organic carbon and both bulk density and Ksat for the two treatments?

A13.16. This is a very good point. Thank you for the suggestion.

[Figure]

As the above figures nicely illustrate, there is no relationship between organic carbon concentration and bulk density visible in Chitedze. However, in Makoka we can see two distinguishable patterns between the two treatments. In the control, the carbon concentration does not vary with bulk density. In the agroforestry treatment, however, we can see a pattern of decreasing bulk density with increasing carbon content.

The correlation between organic carbon and Ksat is more difficult as we have the data from exactly the same location only for some of the Makoka samples.

[Figure]

Also, here we can see quite a clear differentiation between the two treatments, mostly arising from the higher carbon concentration in the agroforestry. The range of Ksat values is slightly shifted towards higher values, however, the differences between the treatments are not significant.

Figure 5: Label (a) for bulk density, (b) for porosity, and (c) for Ksat on the top of the figure (i.e., Figure 5), and explain them in the figure caption. This will enable you to cite them after the end of the results presentation.

What is the need to show the result in big and small cylinder and what is the implication?

This should be clearly explained in the caption that all the depth ranges are independent like 0-5, 6-15, 16-25 cm.

A13.17. Thank you for the feedback on figure 5. We will add labels as suggested for the different variables. We understand that it might appear confusing at first with the different cylinder sizes. We are showing both data sets to not compromise on sample sizes or measurement results because not all analyses were conducted for the small cylinders, however, we have a larger sample size of the small ones. We will adapt the depth ranges in the new version of the manuscript as explained in answer in A4.1 in regard to sampling depth naming.

Line 292: Sometimes, measurement values from the laboratory and the field are presented in the Results section for example, retention curves (tension and soil water content) but there is no explanation or statistical comparison between the two sets of measurements. Since the methods used are standardized, it is unclear why repeated measurements were made, especially when the study's focus is not on evaluating these methodologies. This presentation is confusing (see Figure 6 results).

Which one is correct, then? What is the need for repeated measurements and their presentation in the article? Your topic is not about evaluating the methodology of the two systems, this is very confusing!

A13.18. Thank you for pointing out that this may appear confusing to the reader. The repeated measurements of different types of samples are explained in A13.17. As suggested in A4.1, we will add a table to the manuscript to give a better overview of the samples taken and analyses conducted. Regarding the soil water retention curves, we experienced that field and laboratory retention curves never show exactly the same and both contain slightly different information and advantages and disadvantages. Studies on this topic explain differences so far based on hysteresis effects and representativeness, however, don't have developed an elaborate concept (e.g. Wessolek et al., 1994, Pachepsky et al., 2001, Basile et al., 2003, Iiyama, 2016). Both, laboratory and field retention curve, depict the relationship between soil water content and matric potential. The advantages of the field curves are that they are measured directly in the field under "real"/field conditions and not under controlled laboratory conditions. They are also capable of showing changes over time of the water content-tension relationship, however, not under equilibrium. One critical disadvantage is that it is two separate devices that don't measure in exactly the same location and have difference reference volumes. The laboratory retention curves are measuring changes in exactly the same sampling volume and under controlled equilibrium conditions, which increases accuracy immensely. It describes the relationship rather as a static characteristics and does not include temporal changes.
For the outlined reasons, we found both approaches can provide interesting information.

Wessolek, G., Plagge, R., Leij, F.J., van Genuchten, M.T., 1994. Analysing problems in describing field and laboratory measured soil hydraulic properties. Geoderma 64, 93–110. https://doi.org/10.1016/0016-7061(94)90091-4

Pachepsky, Y., Rawls, W.J., Giménez, D., 2001. Comparison of soil water retention at field and laboratory scales. Soil Sci. Soc. Am. J. 65, 460–462. https://doi.org/10.2136/sssaj2001.652460x

Basile, A., Ciollaro, G., Coppola, A., 2003. Hysteresis in soil water characteristics as a key to interpreting comparisons of laboratory and field measured hydraulic properties. Water Resour. Res. 39, 1–12. https://doi.org/10.1029/2003WR002432

Iiyama, I., 2016. Differences between field-monitored and laboratory-measured soil moisture characteristics. Soil Sci. Plant Nutr. 62, 416–422. https://doi.org/10.1080/00380768.2016.1242367

Line 294: What do you mean?

A13.19. We were referring two the field soil water retention curves overlapping less and being shifted further apart from one another than the laboratory curves. We will reformulate the sentence in the manuscript to clarify: "The field SWRC from different depths are, however, further apart from one another (e.g. in Makoka in the gliricidia plot at depth 0.05 and at 0.25) compared to the laboratory SWRC."

Figure 6. The figures contain many values for different hydrological characteristics but lack labels for each panel. Please add labels such as Soil Water Content (a) Chitedze, (b) Makoka, and PAW (c) Chitedze, (d) Makoka at the top of Figure 6, and explain them clearly in the figure caption. This will enable you to refer to them easily after presenting the results.
Write the title and unites of the measurement.

A13.20. Thank you for suggesting improvements for this figure. We will also add here labels to the specific subplots and use them to refer to the subplots in the main text.

Line 295: What is the reason and the implication from the point view of water availability to the plant?

A13.21. Thank you for the question. A steeper soil water retention curve indicates that the plant-available range of water content in the soil is smaller, because the water contents at field capacity and wilting point lie closer together. This can be seen in figure 6, where the plant-available water content of the laboratory soil water retention curves is shown.

Sometimes the result for one treatment is high at one site but low at the other. For example, in line 298, in Makoka, PAW in the control showed "slightly higher values" compared to the AFS treatment; however, in Chitedze, higher PAW values were recorded in the AFS treatment. There is no reasoning or explanation provided for these contrasting results throughout the Results section.
What is generally the implication of the result? in one site AFS is "greater" than control, vice verse in the other site?

A13.22. In the results section we are only presenting the results from our data avoiding any interpretation. We think it is easier to follow along our argumentation if we first plainly present and describe the data without any judgement and leave the interpretation to the discussion section. Therefore, we would keep this section as it is without including any implications.

Line 302: If the real situation on the ground is steeper slope, what is the need to measure under laboratory conditions? At the end of the day, the results from the laboratory do not represent the real values of the sites.

A13.23. We explain why we are looking at both curves in our response to the comment on line 292.

Line 304: You already mentioned this on the method section if it still not you should explain them there or under the figure cation.

A13.24. Thank you for your suggestion. It is correct that we mentioned this already in the method section. We wanted to include it in the results again as a short reminder in case the reader has forgotten or not read the method section. We will do it as suggested and add the sentence to the caption of figure 7 and remove it from its current location.

Line 305: Did you check the antecedent moisture content and its relationship with overall water content and matric potential?

A13.25. Thank you for this question. The text paragraph you are referring focuses on the overall time series patterns and trends. In this context, it does not really make sense to talk about antecedent soil moisture. We did, however, check the antecedent soil moisture when analysing the precipitation events. In this case, antecedent soil moisture can have in general an effect on hydraulic conductivities and therefore on the response of the sensors registering an increase in water content to rainfall. We did not report these values as they did not any specific/new knowledge to the analysis and we wanted to avoid adding even more data to the already elaborate study.

[Figure]

Line 308: Doesn't it imply the effect of external factors?

A13.26. This question is not quite clear to us what. We are looking at the full timeseries in this paragraph. On average, it is common that the topsoil is over long periods drier because of evapotranspiration and the deeper soil is wetter.

Line 312. What is the reason for the strong drying after rainfall, even in the gliricidia treatment, based on your observations or experience? Could the difference in soil water content between the sites be due to variations in geology or parent material?

A13.27. Thank you for these questions regarding the rainfall events. We understand that we did not clearly elaborate on this in section 4.3, that's why we will go through it

again and rewrite it to clarify this point.
With regard to the parent material's influence on soil water content, soil texture is strongly influenced by parent material and subsequent processes such as weathering and erosion and deposition. The texture itself determines flow and retention rates of the soil, e.g. percolation, water redistribution (in the absence of macropores such as earth worm burrows). However, the differences in soil water content may also arise from difference in precipitation, plant and root abundancy and (micro-)topography.

Figure 7 lacks labeling for P and WC in both the control and gliricidia treatments.
Does it imply that water content at all depth ranges is similar over time?
If available, why don't you show the temperature trend so that we can see the effect of external factors as well?
Missing time series value for all depth ranges or missing for all time series? It is not clear we see one line trend value over time, is it?

A13.28. Thank you for the recommendations to improve figure 7. As with the other figures we will include labels for the panels and refer to them in the text where possible/needed. Also, as suggested before we will add a sentence explaining why there is only one soil water content timeseries visible for the control plot in Chitedze (only the topsoil sensor worked and data from the depths below 10 cm are not available). This should solve the remaining questions regarding this figure.

Line 328: what does it mean?

A13.29. This means that the shortest event lasted one hour and the longest 44 hours. The other events recorded have durations somewhere between one hour and 44 hours (table A3).

Line 335: The statement 'the water slowly percolated downwards in the gliricidia plot as demonstrated by the sequential storage increases with increasing depth', does this truly reflect real field conditions? I have doubts because real-world soil water dynamics are often more complex due to factors such as soil heterogeneity, lateral flows, plant activity, and varying environmental conditions. Therefore, relying solely on this observation may not fully capture the actual behavior of water movement. Additionally, the implications of this for shallow- and deep-rooted plants should be elaborated. Furthermore, the relevance of these findings to the specific climatic zone and potential supplementary irrigation strategies should also be discussed.

A13.30. Thank you for this elaborate comment. It is true that soil water dynamics are governed by many factors including soil heterogeneity, vertical and lateral flows, plant dynamics and other environmental conditions such as the weather.
In this specific example, we are discussing water flow after strong rainfall events and observed a sequential increase in soil water storage during that time. This means we have a direct water input from above into the topsoil. Due to the

absence of strong topographic gradients, it can be assumed that the increases in water content in depth after depth are very likely occurring from the rain water being drawn downward by gravitational forces. Strong lateral flows are more common on slopes or during dry conditions if other forces are stronger than the gravitational potential, e.g. suction by plants.

We were not the first ones to use sequential responses of soil moisture sensors as a proxy for differing between matrix flow and preferential flow, e.g. Demand et al. (2019), Branger and McMillan (2020) and Araki et al. (2022).

Demand, D., Blume, T., Weiler, M., 2019. Spatio-temporal relevance and controls of preferential flow at the landscape scale. Hydrol. Earth Syst. Sci. 23, 4869–4889. https://doi.org/10.5194/hess-23-4869-2019

Branger, F., McMillan, H.K., 2020. Deriving hydrological signatures from soil moisture data. Hydrol. Process. 34, 1410–1427. https://doi.org/10.1002/hyp.13645

Araki, R., Branger, F., Wiekenkamp, I., McMillan, H., 2022. A signature-based approach to quantify soil moisture dynamics under contrasting land-uses. Hydrol. Process. 36. https://doi.org/10.1002/hyp.14553

Line 376. Generalizing the results as 'high' by what index did you classify 'high' and 'very high'? I am also surprised that there was no statistical difference in pedogenic oxide values between the two treatments, yet it was reported that these contribute to the potential structural benefits typically attributed to increased SOM in AFS treatments.

A13.31. The amount of pedogenic oxides is not influenced by treatment but by parent material. The soil C input due to treatment (treatment effect) influences the structure formation. This effect can be stronger when pedogenic oxide values are higher (Kirsten et al., 2021). We will though rephrase the above sentence in Line 376 into "The concentrations of pedogenic oxides (Al, Fe) at Makoka and Chitedze also likely contribute to…"

Kirsten, M., Mikutta, R., Vogel, C., Thompson, A., Mueller, C.W., Kimaro, D.N., Bergsma, H.L.T., Feger, K.-H., Kalbitz, K., 2021. Iron oxides and aluminous clays selectively control soil carbon storage and stability in the humid tropics. Sci. Rep. 11, 5076. https://doi.org/10.1038/s41598-021-84777-7

**14. Discussion**

Figures are misplaced for example, Figure 10 is placed inside the discussion section instead of the appropriate section.

Sometimes unreasonable generalizations are made, even though the facts are only presented in the Results section. For example, in line 356, it states that the soil at the

Chitedze site already has a high carbon saturation level. Did you calculate the Carbon Saturation Index (CSI), which is the ratio of observed SOC to the estimated maximum SOC storage capacity, or use other metrics such as the Mineral-Associated Organic Carbon to Clay + Silt Ratio (MAOC/CS)? In tropical climates, reaching carbon saturation is more challenging than in temperate regions due to higher microbial activity.

A14.1. Thank you for this comment. We discussed in line 356 that soils with high C saturation tend to be closer to their theoretical C saturation point and are therefore less efficient at storing extra C, as written in Stewart et al. (2008). The agroforestry measure in Chitedze was considerably less efficient compared to the site at Makoka and showed no response to C-input measures. Our results indicate that a higher C-saturation at Chitedze compared to Makoka might be one of the reasons for less C accumulation at Chitedze after legume input. We expressed the assumption that soil at the Chitedze site is closer to its C saturation point. It was not meant as generalization and we will clarify that by rephrasing the text passage in the manuscript.
We did not calculate the carbon saturation index nor the mineral-associated organic C to clay and silt ratio as the ratio's effectiveness is debated. We determined though C in soil density fractions where SOM is separated physically into fractions of varying stability as described in line 55-57 to enlighten the proportion of C that is bound to mineral-associated C fraction.

Line 429: Go to the point "In this section we discuss" very weird!

A14.2. Thank you for pointing this out. We will remove the sentence from the manuscript.

Line 432: What is this? was that your hyphothesis?

A14.3. Thank you for the question. This is not an initial hypothesis or research question of the study. It is a follow-up expectation based on the results of the soil hydraulic properties. We will rephrase the sentence to clarify this: "Based on the observed soil hydraulic properties, i.e. similar Ksat values between treatments in Chitedze, we expected similar infiltration responses to rainfall events at both locations. Based on different Ksat values in Makoka, we expected to see faster and more infiltration visible in stronger AFS sensor responses." We will remove the following sentence "However,...treatments" to put more focus on the treatment effects.

Line 438: What is the reason and implication?

A14.4. Thank you for the question. We give an explanation a few lines further down (line 441). One possible reason for the sensors to register less water in the gliricidia intercropping is interception of water by the maize and gliricidia leaves, which cover a larger surface in the gliricidia intercropping than in the control plot, leading to a larger water storage on the leaves.

Line 441: ????? not clear? They are supposed to be together!

Line 441: By what? by the gliricidia plots?

A14.5. We will try to improve our explanation regarding the interception in the gliricidia intercropping compared to the control plot: "One possible reason for the sensors to register less water in the gliricidia intercropping is interception of water by the maize and gliricidia leaves. In the gliricidia intercropping plot the branches and leaves of the gliricidia (when not freshly cut down to the base) cover parts of ground, thereby impeding rainfall to reach the soil surface directly. Additionally, the maize plants appeared more abundant and with larger leaves in the gliricidia intercropping plots compared to the control, which increases to the potential water storage capacities on the leaves in these plots. The interception storage needs to be exceeded by rainfall for water to reach the soil surface."

Line 444: What do mean healthier? larger with what evaluation?

A14.6. Thank you for this question. We did not evaluate plant health and size systematically. We noted down the heights of maize plants in the vicinity of our monitoring equipment and took photos of the surrounding, both indicating that the plants were larger and appeared healthier in the intercropping plot compared to the control. We are not investigating this topic, but plainly observed the situation in the field at the moment we were at the place. We will remove the sentence comparing maize plants in Chitedze with Makoka but focus only on the treatment differences at each location. We will clarify this in the manuscript: "The maize plants appeared larger and more less crop failures occurred in the intercrop than in the control plot."

445-450: In the gliricidia treatment, water movement down the soil profile is context-dependent, particularly influenced by soil texture. What are the implications of this?

A14.7. Thank you for this interesting question. We can see that we didn't discuss this point detailed enough. In the revised manuscript, we will elaborate on the treatment effect with regard to texture and its implication for water movement.

Line 459: Surprisingly, you don't have clay, sand, and silt data for both treatments at the Makoka site. How can you base your reasoning on this?

A14.8. Thank you for pointing out the missing data. We do have textural data for Makoka. Unfortunately, they somehow slipped from the table. As mentioned above we will add them to the manuscript.

Line 459: Would this also be related to the installation of the top sensor in disturbed ridge soil?

A14.9. Thank you for this question. First of all, in general water moves downward through the soil profile and therefore needs more time to reach deeper sensors. The

travelling time depends on the soil structure (macropores) and texture (sand vs clay fractions). Ridges may favour this because they potentially consist of more loosely packed soil with higher infiltration capacities, which reduces the travelling time of the water.

Line 490: This conclusion is difficult to accept because it appears to be based on simple observation without a proper sampling procedure for plant growth and health attributes. Please review how samples are taken with replicates in experimental plots in other published studies.

A14.10. The aim of our study was not to sample for plant growth and health attributes. This is just a small observation on the site, that we noticed while being there and which we use as a possible explanation. Of course, this would have to be tested and confirmed in future studies or related to studies who have conducted this type of field work.

Line 493: What exaltedly?

A14.11. Thanks for pointing this out. We meant to say that the improved plant growth in Chitedze may be attributed to the generally more favourable growing conditions at this site, including higher C and N contents and a higher CEC. We will rephrase the sentence to clarify the meaning: "The improved growth in Chitedze may be attributed to the generally more favorable growing conditions at this site, including higher C and N contents, higher CEC values and therefore possibly more effective nutrient recycling."

Line 498: Does this mean that the effect of Gliricidia residue input on CEC is site-context dependent?

A14.12. Thank you for this question. We were not heading on site specific responses of Gliricidia residue input. As CEC in soil mainly depends on clay and soil organic matter it is not surprising that at Chitedze CEC values correspond with consisting C contents at Chitedze that did not change in the Gliricidia treatment.

**15. Conclusion**

Line 511: "In maize and gliricidia intercropping, we saw a clear treatment effect on soil nutrient and C contents as well as on soil structure." However, regarding the treatment effect, no statistical differences were observed between treatments at either site. You simply label the values as 'high' and 'low' based on the numbers, which is not scientifically valid. Moreover, the treatment effects were not consistent across both sites for all parameters. Please review your tables and figures.

A15.1. Thank you for this comment. As discussed in answer A7.1, we will add the significance tests to the related comparisons and revise the text to emphasize the focus on the treatment effect and away from the site comparisons. We will specify

the sentence to illustrate where precisely we saw significant treatment effects. "In the maize and gliricidia intercropping, we saw a clear treatment effect on soil nutrients and carbon contents as well as on soil texture for one of the two sites."

Line 511: Your work is not a long-term experiment, but it was conducted on a long-term experimental plot.

A15.2. Thank you for pointing this out. We agree that our formulation was misleading here. We will change the sentences as suggested by the reviewer.

Line 518: "In the context of climate change, we saw that some of the envisioned challenges for agriculture in Southern Africa can be alleviated with the adoption of AFSs under certain site conditions." However, your research does not consistently support this conclusion. How did you arrive at this statement?

A15.3. Thank you for noticing this. We understand that the formulation is not ideal. We will rephrase the sentence to clarify our point: „In our example of long-term experiments of maize and gliricidia intercropping, we could show that some of the challenges envisioned for agriculture in South Africa by climate change may be alleviated by agroforestry systems under certain site conditions. One of the two sites, showed clear positive treatment effects such as a more stable soil structure as a consequence of the sustainably supplied organic matter which would potentially be less susceptible to soil erosion. Further, an improved nutrient availability could sustain higher and more stable yields. These clear treatment effects were, however, not found in the second site, highlighting that each agroforestry system needs to be targeted to the individual site conditions. In any case, adding gliricidia at the second site did not lead to any disadvantages in terms of soil water dynamics."